# Evolution of a flipped pathway creates metabolic innovation in tomato trichomes through BAHD enzyme promiscuity

Pengxiang Fan [1], Abigail M. Miller [1,4], Xiaoxiao Liu[2], A. Daniel Jones [1,2] & Robert L. Last [1,3]

Plants produce hundreds of thousands of structurally diverse specialized metabolites via multistep biosynthetic networks, including compounds of ecological and therapeutic importance. These pathways are restricted to specific plant groups, and are excellent systems for understanding metabolic evolution. Tomato and other plants in the nightshade family synthesize protective acylated sugars in the tip cells of glandular trichomes on stems and leaves. We describe a metabolic innovation in wild tomato species that contributes to acylsucrose structural diversity. A small number of amino acid changes in two acylsucrose acyltransferases alter their acyl acceptor preferences, resulting in reversal of their order of reaction and increased product diversity. This study demonstrates how small numbers of amino acid changes in multiple pathway enzymes can lead to diversification of specialized metabolites in plants. It also highlights the power of a combined genetic, genomic and in vitro biochemical approach to identify the evolutionary mechanisms leading to metabolic novelty.

[1] Department of Biochemistry and Molecular Biology, Michigan State University, East Lansing, MI 48824, USA. [2] Department of Chemistry, Michigan State University, East Lansing, MI 48824, USA. [3] Department of Plant Biology, Michigan State University, East Lansing, MI 48824, USA. [4] Present address: Department of Molecular Biology and Genetics, Cornell University, Ithaca, NY 14853, USA. Correspondence and requests for materials should be addressed to R.L.L. (email: lastr@msu.edu)

Plants produce large numbers of structurally diverse specialized metabolites[1], with individual classes typically restricted both taxonomically and spatiotemporally[2,3]. The key families of enzymes involved in producing these metabolites, e.g., cytochrome P450-dependent monooxygenases, terpene synthases, and BAHD acyltransferases, show strong signs of diversification over the course of land plant evolution[4], giving rise to hundreds of thousands of structurally diverse plant metabolites with roles ranging from pollinator and symbiotic interaction to pathogen and herbivore defense[1,5–7]. These enzymes exhibit signs of rapid evolution compared with those of core metabolism[8], and a variety of factors are associated with this lability. For example, specialized metabolic enzymes are encoded by multigene families, and genetic redundancy creates the potential for evolution of novel regulation or enzymatic activities with reduced negative impacts on fitness[9]. In addition, the cell and tissue specificity of specialized metabolic pathways is conducive to changes in enzyme activities without causing adverse fitness effects. Finally, enzyme promiscuity—the ability to utilize multiple structurally related substrates—is associated with the ability of specialized metabolic enzymes to evolve rapidly[10–13]. Changes in one enzymatic activity influence its ability to use substrates produced by other enzymes, and in turn generate products that are substrates for other enzymes. Understanding the evolution of pairs or sets of enzymes can contribute to our knowledge of catalysis and inform novel metabolic engineering strategies.

Several characteristics make the acylsucrose biosynthetic network in cultivated tomato (Solanum lycopersicum) and its wild relatives an exemplary system for understanding biosynthetic pathway evolution. Acylsugars are of interest as natural plant pest control agents, with documented direct protection from fungal pathogen[14] and herbivore[14–17] attack, and indirect protection through tri-trophic interactions[18,19]. The core network in cultivated tomato is relatively simple, with four glandular trichome-expressed BAHD (BEAT, AHCT, HCBT, DAT[20,21]) class acylsucrose acyltransferases (ASATs) catalyzing consecutive reactions[22]. In recent studies, ASATs from Solanaceae species distantly related to the tomato group of Solanum—Petunia axillaris and Salpiglossis sinuata—were characterized, and these create acylsucroses with structures different from those of wild and cultivated tomatoes[23,24]. In cultivated tomato, the entire acylsugar biosynthetic pathway can function in a single test tube with addition of sucrose and acyl-CoA substrates to the four enzymes, permitting facile characterization[22,25]. Despite the compact pathway structure, these enzymes produce diverse tri- and tetra-acylated products in cultivated tomato, due in part to the acyl-CoA substrate promiscuity of the second and third enzymes in the pathway[22,26–28]. Study of wild tomato relatives in the Solanum revealed examples of enzyme and product diversification compared with the domesticated species[27–30]. For example, populations of S. habrochaites from northern Ecuador lack activity of ASAT4, the last enzyme of the pathway, resulting in loss of pyranose ring $R_2$ acetylation[29]. Interspecific differences in ASAT2 and ASAT3 acyl-CoA substrate preference cause acyl chain variation at the pyranose $R_4$ and $R_{3'}$ positions, respectively[22,27]. Despite these differences, most wild and cultivated tomato acylsucroses contain a single acyl chain on the furanose ring at the sucrose $R_{3'}$ position ("-F type" acylsucroses, see Fig. 1a)[27,29].

In contrast, S. habrochaites and S. pennellii accumulate an unusual set of triacylated sucroses decorated only on the six-member pyranose ring ("-P type" acylsucroses, see Fig. 1a)[28,30]. We found that these species produce ASAT3-P enzyme, which is a variant that uses a monoacylated acylsucrose acceptor substrate, rather than the diacylated substrates used by S. lycopersicum type Sl-ASAT3 activity[27]. However, this work left open the question of

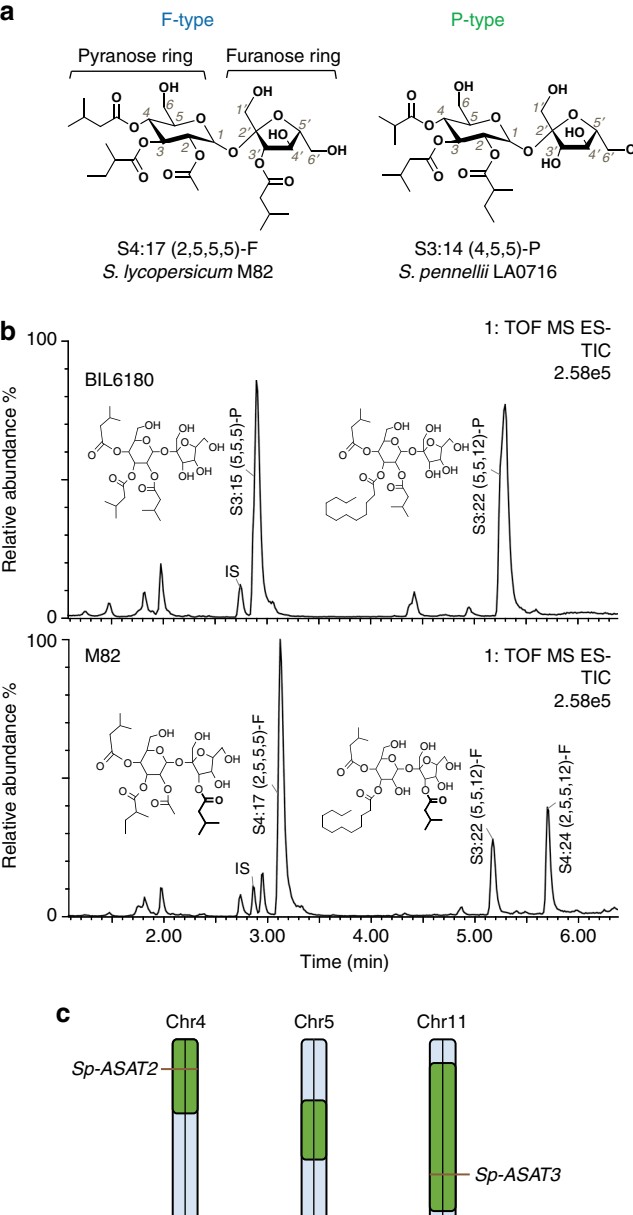

**Fig. 1** The recombinant line BIL6180 produces abundant acylsucroses with all acyl chains on the pyranose ring. **a** Structures of representative F- and P-type acylsucroses purified from S. lycopersicum M82 and S. pennellii LA0716, respectively. The F-type acylsucrose has acylations on the pyranose and furanose rings, with a single acyl chain at $R_{3'}$ position. The P-type acylsucrose has three acyl chains on the pyranose ring, with no furanose ring acylation. **b** BIL6180 produces abundant P-type acylsucroses, while S. lycopersicum M82 accumulates mainly tri- and tetra- F-type acylsucroses in the trichomes. IS: internal standard. The two total ion chromatograms are scaled to the same intensity, where the major peak (S4:17-F) in M82 has the maximum signal (100% scale). **c** BIL6180 has three chromosome regions from S. pennellii (green) introgressed into S. lycopersicum M82 background (blue). The introgressions on chromosomes 4 and 11 include Sp-ASAT2 and Sp-ASAT3, respectively

what enzyme converts the diacylated ASAT3-P product to the final triacylated P-type acylsucrose, which has all three substitutions on the pyranose ring.

In the current study, we address this question using a combination of genetic and transgenic plant approaches along with site-

directed mutagenesis and in vitro biochemistry. We identify a cultivated tomato line that produces P-type acylsucroses and found that the concerted action of two regions of the *S. pennellii* LA0716 genome causes this novel phenotype. These regions harbor the *Sp-ASAT2* and *Sp-ASAT3* genetic loci, and P-type acylsucrose accumulation results from epistasis between these loci. The combination of these alleles is necessary and sufficient to produce the pyranose ring acylated products in vivo. Biochemical analysis demonstrates that the P-type acylsucroses result from reaction of Sp-ASAT3 followed by Sp-ASAT2, which is the reverse order of action compared with that of the cultivated tomato enzymes. In vitro mutagenesis reveals that a small number of amino acid changes in each enzyme is sufficient to switch the activities between the F- and P- acylsucrose pathways. Investigation of these pathways in other tomato species allows inference of the evolutionary events giving rise to divergence of F- and P-type acylsucroses. This work demonstrates multiple evolutionary mechanisms in biochemical pathway evolution, including gene duplication and loss, amino acid substitution, and how emergence of pathway epistatic interactions restructured a specialized metabolic pathway and led to metabolic product innovation.

## Results

**Two *S. pennellii* loci influence acylsucrose acylations.** Single chromosomal introgressions from *S. pennellii* LA0716 cause dramatic changes in the types of specialized metabolites that accumulate in cultivated tomato trichomes[22,26,27,30–34] and other tissues[35]. The availability of Backcrossed Inbred Lines (BILs), containing multiple *S. pennellii* chromosomal regions in individual *S. lycopersicum* lines[33,35,36] provides an opportunity to identify epistatic interactions that influence the types of metabolites that accumulate. We analyzed trichome acylsugar metabolites of 257 BILs by liquid chromatography-time of flight mass spectrometry (LC-ToF MS) to look for phenotypes not observed in single chromosomal introgression lines.

One BIL line—referred to as 6180 in this report—was found to produce an acylsugar profile absent from *S. lycopersicum* M82 or single introgression lines (ILs) previously screened[32] (Fig. 1b). BIL6180 exclusively accumulated detectable triacylsucroses with all three acylchains on one ring, as revealed by positive mode mass spectrometry (MS) (Supplementary Fig. 1a). These resemble the acylsugars produced in *S. pennellii* and *S. habrochaites*[27] and the minor acylsugar peaks produced in IL4-1[22]. NMR analysis of two of the major BIL6180 triacylsucrose peaks, namely S3:15 (5, 5, 5) and S3:22 (5, 5, 12) (note that "S" refers to a sucrose backbone, "3:15" indicates three acyl chains with total fifteen carbons and the length of each acyl chain is shown in the parentheses), showed that they were acylated exclusively on the six-membered pyranose ring at the $R_2$, $R_3$, and $R_4$ positions (Supplementary Fig. 1b). This pattern is in contrast to that of the parent *S. lycopersicum*, which accumulates F-type acylsucroses with a single acyl chain at the $R_{3'}$ position of the five-membered furanose ring (Fig. 1a). However, it resembles the acylation pattern of the *S. pennellii* LA0716 acylsucrose S3:19-P (5$^{R2}$, 10$^{R3}$, 4$^{R4}$) previously resolved by NMR[30]. Another similarity between the BIL6180 and *S. pennellii* LA0716 acylsucroses is that both S3:22-P (5$^{R2}$, 12$^{R3}$, 5$^{R4}$) and S3:19-P (5$^{R2}$, 10$^{R3}$, 4$^{R4}$) have the long acyl chain—C12 in 6180 and C10 in *S. pennellii* LA0716—at the $R_3$ position, with C4 or C5 chains at $R_2$ and $R_4$ (Supplementary Fig. 1b). Furthermore, each detected BIL6180 acylsucrose peak was composed of multiple isomers differing by branched acyl chain types. Liquid chromatography-based separation resulted in the co-elution of a mixture of S3:15 (5,5,5) structural isomers (Supplementary Fig. 1b), and their NMR

spectra revealed either iC5 or aiC5 at each of the three acylation positions. In addition, two co-eluting S3:22 (5, 5, 12) isomers (Supplementary Fig. 1b) differed by the presence of iC5 or aiC5 acyl chains at the $R_4$ position. Taken together, these results indicate that BIL6180 accumulates pyranose ring acylated triacylsucroses lacking furanose ring acylation, which are similar to those found in the *S. pennellii* LA0716 parent (Fig. 1b). The similarity between the acylsucroses from BIL6180 and *S. pennellii* LA0716 led us to hypothesize that BIL6180 could be utilized to characterize the *S. pennellii* acylsucrose biosynthetic pathway.

**Sp-ASAT2 and Sp-ASAT3 make P-type acylsucroses.** BIL6180 contains three *S. pennellii* introgressions, including regions on chromosome 4 and 11 that contain *Sp-ASAT2* and *Sp-ASAT3*, respectively (Fig. 1c). These are orthologs of the *S. lycopersicum* ASAT genes *Sl-ASAT2* and *Sl-ASAT3*, which encode enzymes catalyzing the second and third steps of the cultivated tomato acylsucrose biosynthetic pathway[22,27]. This led to the hypothesis that Sp-ASAT2 and Sp-ASAT3 contribute to biosynthesis of P-type acylsucroses in *S. pennellii* LA0716. To determine whether the *S. pennellii* loci that led to P-type acylsucrose accumulation in BIL6180 are dominant or recessive, we crossed homozygous BIL6180 with the parent *S. lycopersicum* M82 and analyzed the leaf acylsugar profiles of the F1 progeny by LC-ToF MS (Supplementary Fig. 2). Both F- and P-type acylsucroses accumulated in the F1 plants, indicating that the loci contributing to F- and P-type acylsucroses are co-dominant, which led us to design transgenic experiments to test our hypothesis.

We engineered *S. lycopersicum* M82 plants by simultaneously expressing the *Sp-ASAT2* and *Sp-ASAT3* coding regions driven by the type I/IV trichome specific promoters[22,27] of *Sl-ASAT2* and *Sl-ASAT3*, respectively (Fig. 2a). As predicted, the P-type acylsucroses S3:15 (5, 5, 5)-P and S3:22 (5, 5, 12)-P were produced in the regenerated transgenic T0 plants, but not in the parental M82 (Fig. 2b and Supplementary Fig. 3). Similar to the F1 double heterozygotes (Supplementary Fig. 2), the F-type acylsucroses were also detected (Fig. 2b and Supplementary Fig. 3). This result provides strong evidence that *Sp-ASAT2* and *Sp-ASAT3* are sufficient for P-type acylsucrose production in *S. pennellii* and BIL 6180, and led us to characterize their enzymatic activities.

**Reversed reaction order leads to P-type acylsucroses.** To test their role in P-type acylsucrose biosynthesis, we performed in vitro assays using purified Sp-ASAT2 and Sp-ASAT3 proteins. Published work demonstrated that Sl-ASAT2 acylates the monoacylsucrose S1:5 (iC5$^{R4}$) pyranose ring R3 position to produce a diacylsucrose[22]. In contrast, Sp-ASAT2 does not use the monoacylated sucrose as a substrate with any of the acyl-CoA donors[22]. Surprisingly, the *S. pennellii* LA0716 ortholog of the third enzyme in the cultivated tomato pathway—Sp-ASAT3— utilized monoacylated sucrose and iC5-CoA donor substrates to produce a diacylated product with the second chain on the pyranose ring[27]. NMR analysis of the purified diacylsucrose S2:10 (iC5, iC5$^{R4}$) produced by the combination of of Sl-ASAT1 and Sp-ASAT3 revealed that the second chain is acylated at the $R_2$ position of the pyranose ring (Supplementary Fig. 4). Finally, HPLC analysis revealed that the diacylsucrose S2:10 (iC5$^{R2}$, iC5$^{R4}$) made by Sp-ASAT3 did not co-elute with the Sl-ASAT2 product S2:10 (aiC5$^{R3}$, iC5$^{R4}$) (Fig. 3a), supporting the hypothesis that the two diacylsucroses made by Sp-ASAT3 and Sl-ASAT2 are different compounds.

The production of S2:10 (iC5$^{R2}$, iC5$^{R4}$) by Sp-ASAT3 led us to hypothesize that the *S. pennellii* LA0716 ASAT2 ortholog uses this diacylsucrose as an acceptor substrate to produce the P-type

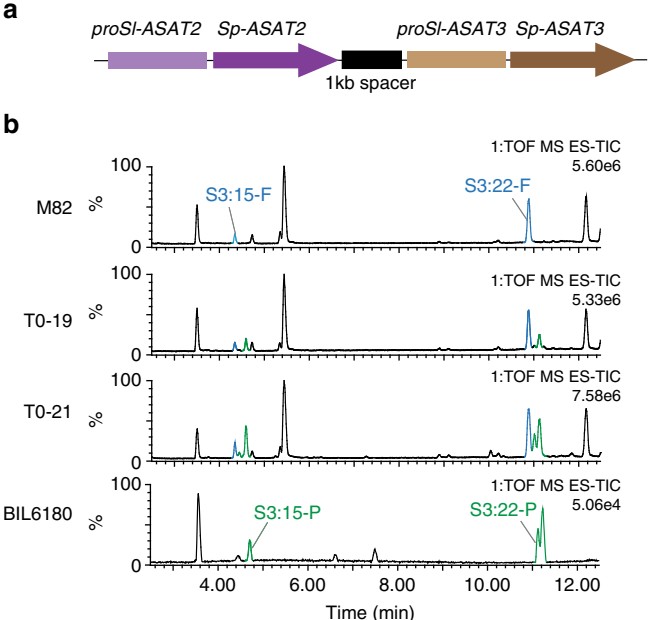

**Fig. 2** Transgenic expression of *Sp-ASAT2* and *Sp-ASAT3* causes trichomes of cultivated tomato *S. lycopersicum* M82 to produce P-type acylsucroses. **a** Schematic representation of the transgenic locus used to express *Sp-ASAT2* and *Sp-ASAT3*, driven by the *Sl-ASAT2* and *Sl-ASAT3* promoters, respectively. A 1 kb spacer separates these genes. **b** T0 first generation transgenic plants accumulate the F-type acylsucroses S3:15-F and S3:22-F, which are produced in M82, as well as the P-type acylsucroses S3:15-P and S3:22-P found in BIL6180. The Y-axes represent abundance of each peak relative to the most abundant ion intensity in each LC-MS run

acylsucroses found in this species and in BIL6180 (Fig. 1b). Indeed, Sp-ASAT2 produced triacylsucroses (Fig. 3b) with the same positive-ion mode MS fragment ion masses (Supplementary Fig. 1a) and chromatographic retention times (Fig. 3b) as the BIL6180 acylsucroses. These results are consistent with the hypothesis that the in vitro products and in vivo metabolites have the same structures. Based on these results we deduce that Sp-ASAT2 adds the third acyl chain to the pyranose ring $R_3$ position, producing the triacylsucroses S3:15 ($5^{R2}$, $5^{R3}$, $5^{R4}$) and S3:22 ($5^{R2}$, $12^{R3}$, $5^{R4}$). The diacylsucrose S2:10 (iC5$^{R2}$, iC5$^{R4}$) is not an acyl acceptor substrate for any characterized cultivated tomato enzyme (Sl-ASAT1, Sl-ASAT2, Sl-ASAT3 or Sl-ASAT4). The combined biochemical evidence argues that *S. pennellii* LA0716 produces triacylsucroses with decorations entirely on the pyranose ring by the stepwise reactions of Sp-ASAT1, Sp-ASAT3 and Sp-ASAT2. Therefore, the *S. pennellii* LA0716 acylsucrose biosynthetic pathway can be drawn side by side with that of cultivated tomato *S. lycopersicum* by reversing the enzyme action order of ASAT2 and ASAT3 (Fig. 3c).

We tested this 'flipped pathway' hypothesis by transgenic RNA interference (RNAi) silencing of *Sp-ASAT3* in BIL6180. We expected greatly reduced acylsugar accumulation in these silenced lines if Sp-ASAT3 catalyzes the second step in the *S. pennellii* LA0716 pathway. This hypothesis is based on past work that showed that monoacylated sucrose S1:5 (iC5$^{R4}$) does not accumulate to detectable levels in *Sl-ASAT2* silenced *S. lycopersicum* M82[22]. Indeed, most of the BIL6180 T0 transgenic lines had reduced total acylsugar levels (Supplementary Fig. 5), with diacylsucroses undetected in the silenced lines. These transgenic plant results reinforce the in vitro enzyme assays showing that Sp-ASAT3 catalyzes the second biosynthetic step of acylsucroses in BIL6180 and *S. pennellii* LA0716 (Fig. 3c).

**Amino acids that contribute to F- and P-type ASATs divergence.** We used a phylogeny-guided approach[22,25] to identify amino acid substitutions responsible for the differences in activities of the orthologous ASAT2 and ASAT3 enzymes in cultivated tomato and *S. pennellii* LA0716. This approach benefited from the observation that both *Sp-ASAT2* and *Sp-ASAT3* share more than 92% identity at the amino acid level with their orthologs in cultivated tomato. In addition, ASAT2 orthologs exist in wild tomato species that fall within the Solanum tomato clade that includes cultivated tomato and *S. pennellii*[37]. We tested the activities of ASAT2 variants from these wild tomato species using S1:5 (iC$^{R4}$) and S2:10 (iC5$^{R2}$, iC5$^{R4}$) as acyl acceptor substrates. As shown in Fig. 4a, the presence of Gly at position 304 correlates with the ability of ASAT2 to use S2:10 (iC5$^{R2}$, iC5$^{R4}$) as a substrate. In contrast, Gln$^{135}$ and Tyr$^{136}$ are found in ASAT2 enzymes that use S1:5 (iC5$^{R4}$) as a substrate (Fig. 4a). In vitro mutagenesis of C304G in Sl-ASAT2 led to the ability to convert S2:10 (iC5$^{R2}$, iC5$^{R4}$) to the P-type triacylsucrose S3:22 (iC5$^{R2}$, nC12$^{R3}$, iC5$^{R4}$) (Fig. 4b). This single amino acid substitution made the enzyme more promiscuous because it retained the Sl-ASAT2 ability to use S1:5 (iC5$^{R4}$) as a substrate (Supplementary Fig. 6). The predicted change in substrate specificity was seen following in vitro mutagenesis of Sp-ASAT2 to generate the mutant with both substitutions H135Q and C136Y: this mutant enzyme gained the ability to utilize S1:5 (iC$^{R4}$) as the acceptor, producing the diacylsucrose S2:10 (aiC5$^{R3}$, iC5$^{R4}$), as did wild-type Sl-ASAT2 (Fig. 4c). Mutation of both amino acids (135 and 136) was required for complete conversion: both single mutants of Sp-ASAT2 H135Q and C136Y showed a residual ability to use S1:5 (iC$^{R4}$) as a substrate (Supplementary Fig. 7).

To assess whether the residues associated with ASAT2 F- or P-type activity have a similar function in other tomato relatives, the F-type ASAT2 from *S. arcanum* LA2172 and the P-type ASAT2 from *S. pennellii* LA1926 were cloned and subjected to in vitro mutagenesis. Indeed, *S. arcanum* LA2172-ASAT2 with substitution C304G gained the P-type ability to acylate S2:10 (iC5$^{R2}$, iC5$^{R4}$) (Supplementary Fig. 8a). Similar to the *S. pennellii* LA0716 Sp-ASAT2 mutant with both residue changes H135Q and C136Y, substitution of P135Q and C136Y in *S. pennellii* LA1926-ASAT2 enabled the enzyme to catalyze the F-type activity by acylating S1:5 (iC5$^{R4}$) (Supplementary Fig. 8b). These results corroborated those obtained by mutagenesis of Sl-ASAT2 and Sp-ASAT2.

We employed a similar approach to identify amino acid residues that shape ASAT3 enzyme activities. A previous study documented 35 *ASAT3* variants from *S. habrochaites* that belong to two clades—F- and P-type—which catalyze furanose and pyranose ring acylation similar to Sl-ASAT3 and Sp-ASAT3, respectively[27]. A survey of amino acids from six Sh-ASAT3 from each enzyme group (Fig. 5a), revealed a correlation between residues at positions 161, 162, and 289 and enzyme types (Fig. 5a). We tested the importance of these amino acids by making the Sl-ASAT3 triple mutant with Y161H, C162S and T289V: this engineered protein gained the ability to acylate S1:5 (iC5$^{R4}$), as seen in P-type ASAT3 enzymes (Fig. 5b). This mutant was promiscuous for acceptor substrate, maintaining the ability to acylate the furanose ring of S2:10 (aiC5$^{R3}$, iC5$^{R4}$) (Supplementary Fig. 6b). Interestingly, each of the three Sl-ASAT3 single-residue mutants gained a partial ability to acylate S1:5 (iC5$^{R4}$), though less well than the triple mutant (Supplementary Fig. 9a). The Sp-ASAT3 triple mutant—with the reciprocal changes H161Y, S162C, and V289T—lost the ability to acylate S1:5 (iC5$^{R4}$) without gaining F-type ASAT3 activity.

Further sequence comparison of the twelve ShASAT3 variants revealed that the residues at position 354, 381, and 382 correlate with enzyme group differences (Fig. 5a). In Sl-ASAT3, these residues are Leu$^{354}$, His$^{381}$, and Pro$^{382}$, whereas in Sp-ASAT3

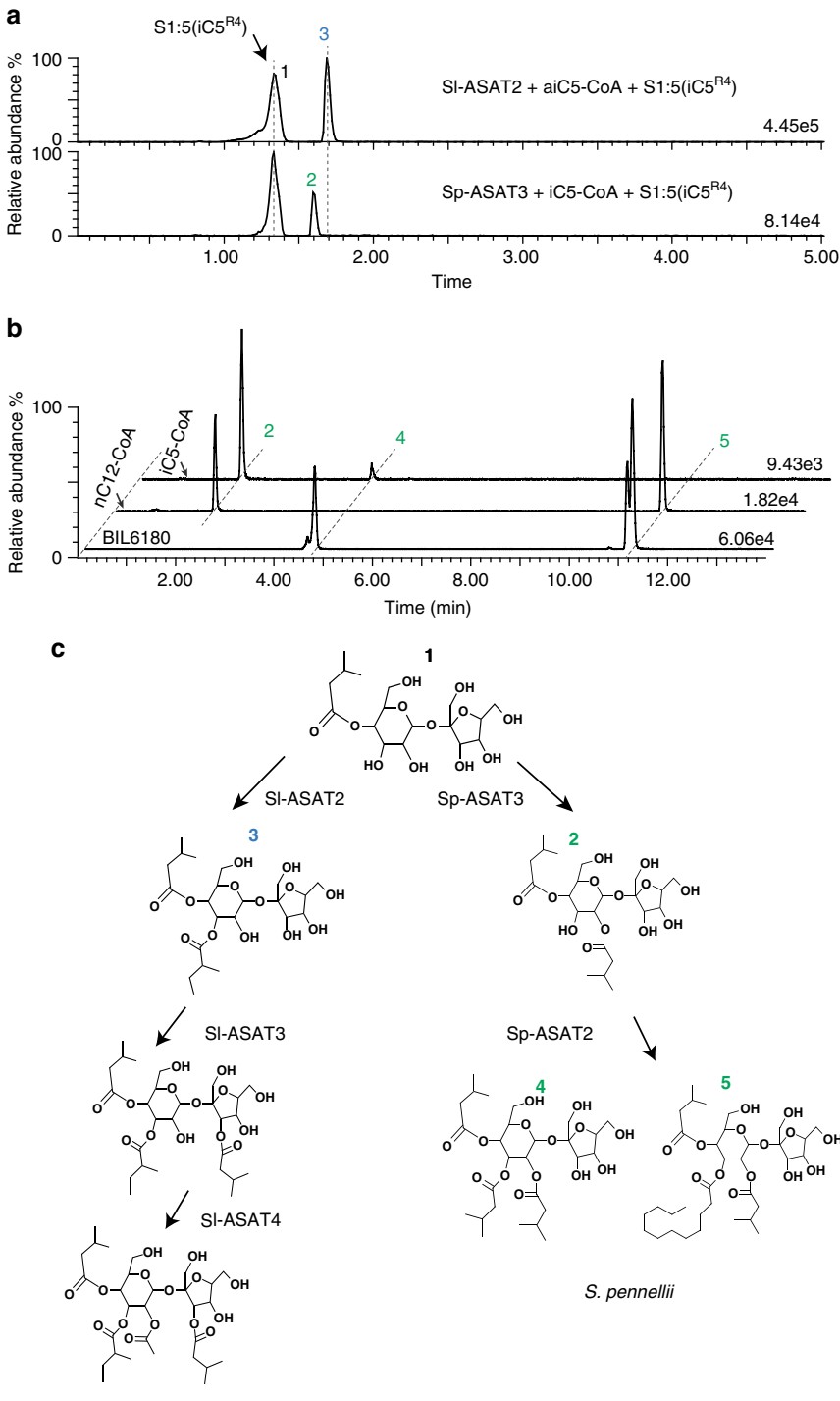

**Fig. 3** Sp-ASAT2 and Sp-ASAT3 produce P-type acylsucroses via a "flipped pathway". **a** *S. pennellii* LA0716 wild tomato Sp-ASAT3 acylates S1:5 (iC5$^{R4}$) and produces a diacylsucrose product (peak 2) with a retention time distinct from that made by cultivated tomato Sl-ASAT2 (peak 3). **b** Sp-ASAT2 converts the S2:10 (iC5$^{R2}$, iC5$^{R4}$) diacylsucrose produced by Sp-ASAT3 (peak 2) to triacylsucroses with the same retention times as those produced by BIL6180 (peaks 4 and 5). "iC5-CoA" and "nC12-CoA" refer to the two different acyl donor substrates used in the two enzyme reactions. **c** The "flipped" acylsucrose biosynthetic pathways in *S. lycopersicum* and *S. pennellii* LA0716. The numbers on top of each compound refer to the chromatographic peaks in **a** and **b**

they are Val$^{353}$ and Ser$^{380}$, with one amino acid deletion adjacent to position 380 compared with Sl-ASAT3. To test whether these residues are associated with the ability of ASAT3 to acylate the furanose ring of S2:10 (aiC5$^{R3}$, iC5$^{R4}$), Sp-ASAT3 was mutagenized to generate the following changes: V353L and S380HP. Indeed, the mutagenized enzyme gained the F-type ability to

acylate S2:10 (aiC5$^{R3}$, iC5$^{R4}$) in vitro (Fig. 5c). Meanwhile, we detected a small amount of F-type activity for Sp-ASAT3 mutants with the single substitutions V353L or S380HP (Supplementary Fig. 9b). While these amino acid substitutions correlate with—and predict—enzymatic activity in the twelve ASAT3 proteins documented in Fig. 5, there are 10 P-type ASAT3 enzymes that

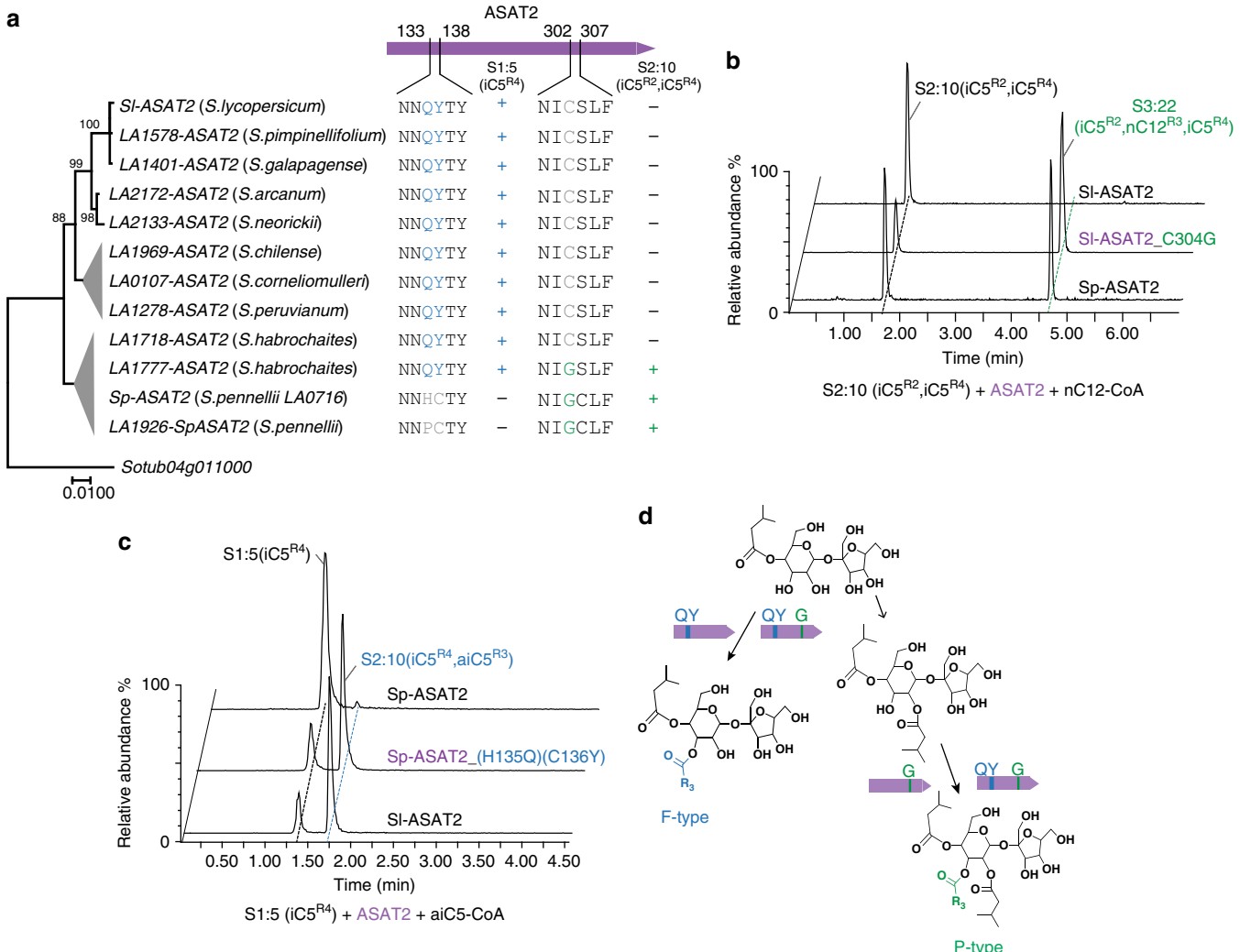

**Fig. 4** Identification of key residues that contribute to switching ASAT2 activities between productions of F- or P-type acylsucroses. **a** Correlation between ASAT2 enzyme activities and amino acid sequences for various tomato species. The maximum likelihood phylogenetic gene tree was obtained using the nucleotide sequences of *ASAT2* variants, with the *ASAT2* homolog (*Sotub04g011000*) in *S. tuberosum* serving as outgroup. The bootstrap values were obtained with 1000 replicates. The nodes with bootstrap values lower than 70 were collapsed. The ability of different ASAT2 to use acyl acceptor substrates S1:5 (iC5$^{R4}$) or S2:10 (iC5$^{R2}$, iC5$^{R4}$) is shown by "+", while "−" indicates no detected enzyme activity. Presence of Q$^{135}$Y$^{136}$ and G$^{304}$ correlate with activity with S1:5 and 2:10 acceptor substrates, respectively. **b** Results of in vitro assays performed with the site-directed mutant enzymes. Sl-ASAT2 with the residue substitution C304G produced the P-type triacylsucrose product that has the same chromatographic retention time with Sp-ASAT2's product using S2:10 (iC5$^{R2}$, iC5$^{R4}$) as the substrate. **c** Sp-ASAT2 with the dual residue substitutions H135Q and C136Y produced the diacylsucrose product that has the same chromatographic retention time with Sl-ASAT2's product using S1:5 (iC5$^{R4}$) as the substrate. **d** Interpretation of the results of parts **b** and **c**, showing the key amino acids associated with F- and P-type ASAT2 enzyme activities

do not follow the Val$^{353}$ and Ser$^{380}$ pattern (gray circles in Supplementary Fig. 12).

Taken together, this combined comparative biochemical and mutagenesis analysis revealed multiple ASAT2 and ASAT3 residues that contribute to the divergence of ASAT2 and ASAT3 enzyme activities, which in turn result in F- or P-type acylsucroses. As expected for amino acids that influence substrate specificity, homology modeling with the trichothecene 3-O-acetyltransferase−acyl CoA complex structure (PDB code 3B2S)[38] predicted that these residues surround the reaction center (Supplementary Fig. 10).

**Acylsucrose pathway divergence in the Solanum tomato clade.** We previously characterized the distribution pattern of F- or P-type acylsucroses in the tomato group[27]. The results described

above led to the hypothesis that the distribution of F- and P-type acylsucroses across this group reflects the divergence of the acylsucrose pathway in the tomato clade driven by *ASAT2* and *ASAT3*. To test this hypothesis, we sequenced the *ASAT2* and *ASAT3* genes of species across this group and determined whether the plant acylsucrose phenotypes can be inferred using *ASAT2* and *ASAT3* genotypes.

In the tomato subclade ranging from *S. lycopersicum* through *S. corneliomulleri*, all characterized ASAT2 proteins in this tomato subclade possessed only F-type enzymatic activities (Fig. 4a). In addition, sequenced ASAT3 orthologs from five species in this subclade all have the F-type activity-associated residues Leu$^{353}$ and His$^{381}$ Pro$^{382}$ (see bottom of Supplementary Fig. 12), which predicts that all of these ASAT3 orthologs have F-type enzymatic activities. The in vitro enzyme activities were tested for three ASAT3 from *S. arcanum* (LA2172-ASAT3_1), *S. huaylasense*

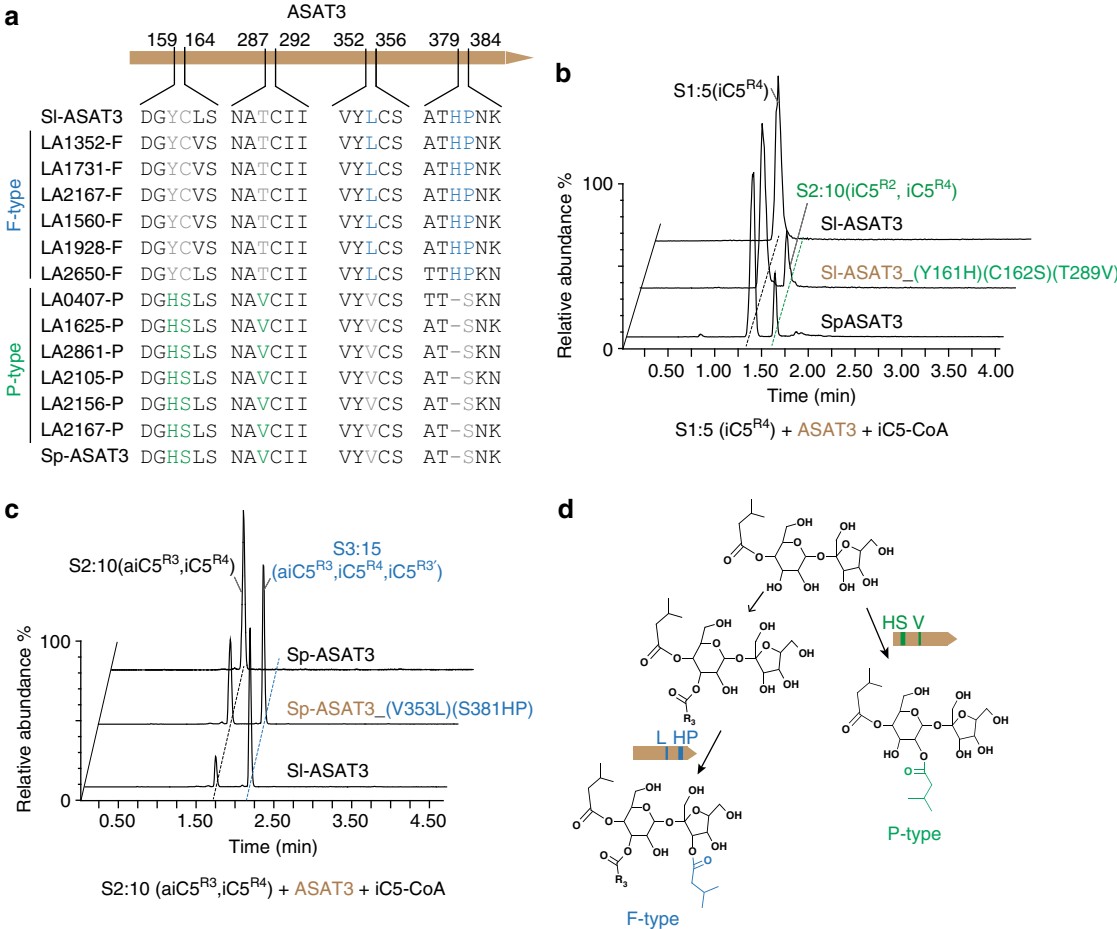

**Fig. 5** Identification of residues that confer P-type activity to Sl-ASAT3 or F-type activity to Sp-ASAT3. **a** Amino acid sequences of ASAT3-P or -F alleles from different *S. habrochaites* accessions revealed residues that correlate with each enzyme type, and these are color coded. Blue amino acids are associated with F-type and green residues with P-type activity. **b** The results of in vitro assays performed with the two parental enzymes and the triple mutant created by site-directed mutagenesis of Sl-ASAT3 enzyme. The Sl-ASAT3 protein with three residue substitutions—Y161H, C162S, and T289V—produced the diacylsucrose product that has the same chromatographic retention time with the product of Sp-ASAT3 using S1:5 (iC5$^{R4}$) as the substrate. **c** The double amino acid position mutant Sp-ASAT3—with V353L and S380HP—produced the triacylsucrose product that has the same chromatographic retention time with the product of Sl-ASAT3 using S2:10 (aiC5$^{R3}$, iC5$^{R4}$) as the substrate. **d** Interpretation of the results of parts **b** and **c**, showing the key amino acids associated with F- and P-type ASAT3 enzyme activities

(LA1364-ASAT3), and *S. peruvianum* (LA1278-ASAT3_1). Indeed, they have the F-type activity, acylating S2:10 (aiC5$^{R3}$, iC5$^{R4}$) (Supplementary Fig. 13, left panel). The combination of ASAT2-F and ASAT3-F alleles predicts that species in this group would only have the F-type acylsucrose biosynthetic pathway (Fig. 6b), matching the previously described F-type trichome acylsucroses of *S. arcanum*, *S. neorickii*, *S. huaylasense*, *S. peruvianum*, and *S. corneliomulleri*[27]. Interestingly, LA1364-ASAT3 and LA1278-ASAT3_1 have residues His$^{161}$ and Ser$^{162}$, respectively; each was associated with the P-type activity from the *S. habrochaites* and *S. pennellii* analysis. Consistent with this observation, these two ASAT3 enzymes are promiscuous, showing a minor P-type activity to acylate the R$_2$ position of S1:5 (iC5$^{R4}$) (Supplementary Fig. 13, right panel).

In contrast to other tomato species, only P-type acylsucroses were detected in *S. pennellii* accessions[27], suggesting that the 'LA0716 biosynthetic pathway' alone is present in these accessions. Indeed, all ASAT3 isoforms isolated from ten *S. pennellii* accessions only have the residues His$^{161}$Ser$^{162}$ and Val$^{289}$, which are associated with P-type activity (Fig. 6a and Supplementary Fig. 12). This suggests that the uniform P-type ASAT3 activity would direct the pathway to produce P-type acylsucroses (Fig. 6b).

In addition, ASAT2 cloned from five *S. pennellii* accessions originating from south Peru (Supplementary Fig. 14) possess the residue Gly$^{304}$, which is associated with P-type activity (Supplementary Fig. 11). However, *S. pennellii* ASAT2 cloned from northern Peruvian accessions (Supplementary Fig. 11) have both F- and P-type activity-associated residues (Supplementary Fig. 14), which predicts that they have hybrid F- and P- type enzyme activities: this combination is reminiscent of *S. habrochaites* LA1777-ASAT2 (Fig. 4a). Although this ASAT2 F-type activity would enable the pathway to branch and make both F- and P-type acylsucroses, the F-type diacylsucrose intermediates are not expected to be acylated further due to a lack of ASAT3 F-type activity (Fig. 6b). This presumably is why no F-type acylsucroses were found in these northern Peruvian accessions[27].

*S. habrochaites* is the only species in the tomato group with accessions demonstrated to accumulate a mixture of F- and P-type acylsucroses[27], providing information regarding the evolutionary relationship of the F- and P- type acylsucroses pathways. Our previous study identified 13 *S. habrochaites* accessions with both the F and P alleles of ASAT3[27]. To extend this study, we further cloned sequences for ASAT2 enzymes from six of the *S. habrochaites* accessions that harbored both *ASAT3-F*

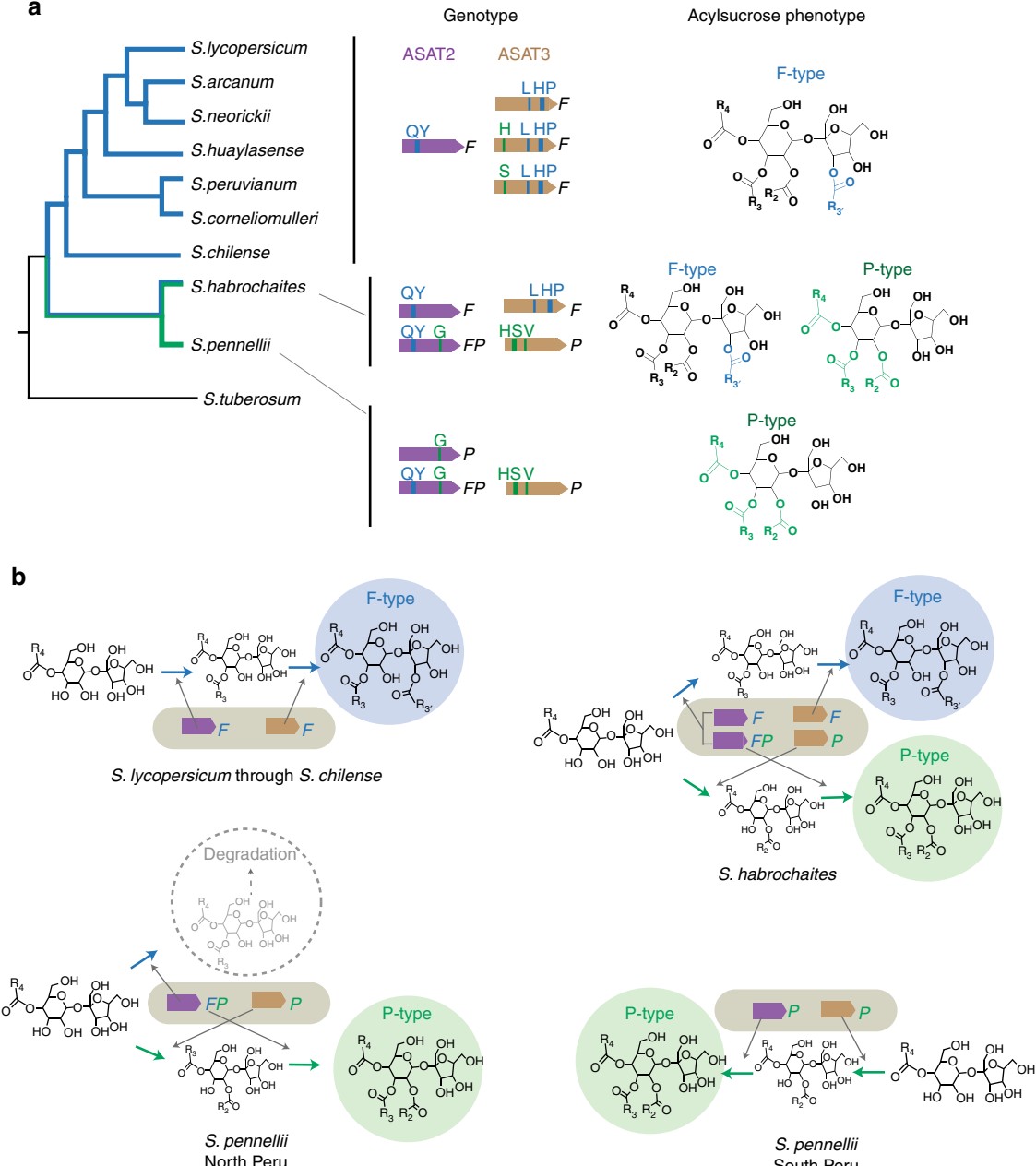

**Fig. 6** Solanum tomato group *ASAT2* and *ASAT3* variant proteins in a phylogenetic context. **a** *ASAT2* and *ASAT3* variants cloned from different wild tomato species are assigned as F- or P- protein types based on residues identified by in vitro mutagenesis. Amino acids associated with F- or P-type activities are in blue and green, respectively. See Supplementary Figs. 11 and 12 for detailed gene phylogenies. The plant trichome F- or P- acylsucrose phenotypes are based on published results. There are NMR structural data for acylsugars from *S. lycopersicum, S. habrochaites* and *S. pennellii*[28,30], while LC/MS data exist for all species listed[27]. **b** Summary of the types of ASAT and product combinations found in the tomato clade. The combination of ASAT2 and ASAT3 enzyme types in different tomato species was inferred based on the activity-associated residues as shown in **a**. Degradation of the dead end diacylsucrose is based on published results for loss of function *S. lycopersicum* ASAT3[27] and characterization of acylsucrose hydrolases in *S. lycopersicum* and *S. pennellii*[30]

and *ASAT3-P*, including accessions LA1772, LA1978, LA2156, LA2650, LA2722, and LA2861. We identified at least two *ASAT2* alleles from each of the selected *S. habrochaites* accessions; in each case, one ASAT2 harbors both F- and P-type residues and the other one only possesses the F-type residues (Supplementary Fig. 11). The combination of ASAT2 and ASAT3 with both F- and P-types suggests the co-existence of F- and P-type acylsucrose biosynthetic pathway in these plants (Fig. 6b), which is consistent with *S. habrochaites* species having the most varied acylsucrose phenotypes across the tomato group described to date[27,29].

## Discussion

Metabolic innovation events can endow an organism with a greater ability to survive in a challenging environment[39,40]. The mixture of specialized metabolites produced in a plant species or group results from assembly of enzymes into metabolic pathways or networks, and evolution of novel products occurs one step at a time. In this process, change in the activity of a single enzyme to produce a new product can lead to a variety of consequences. The simplest productive outcome is that the next enzyme in the pathway is promiscuous and uses the new product as a substrate, converting it to a product that is slightly different from the

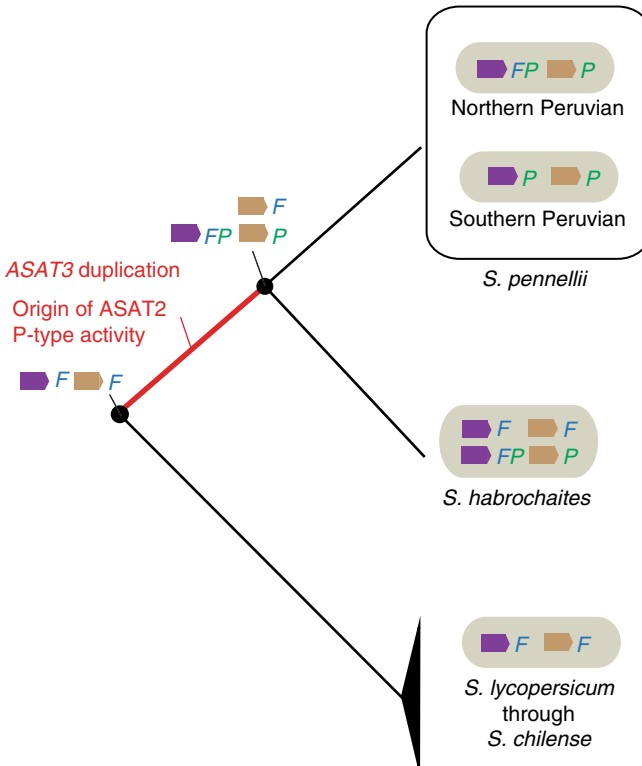

**Fig. 7** Summary of results suggesting that P-type ASAT2 and ASAT3 enzyme evolution occurred after the last common ancestor of the tomato clade of Solanum. Ancestral *ASAT2* and *ASAT3* genotypes were inferred based on the *ASAT2* (Supplementary Fig. 11) and *ASAT3* (Supplementary Fig. 12) gene phylogenies. It suggests that the *ASAT3* duplication event and the origin of ASAT2 P-type activity occurred before the speciation of *S. habrochaites* and *S. pennellii* (the red-colored branch) and away from the lineage containing the cultivated tomato *S. lycopersicum* and its closer relatives

predecessor. A second possibility is that the new product may not be a substrate for any available biosynthetic enzymes, precluding further conversion steps. Alternatively, the new product might be a substrate for an enzyme not normally involved in the predecessor pathway—so called 'silent' or 'occult' metabolism[41]—giving rise to a novel metabolic product. The example described in this paper represents a more complicated case, involving changes in substrate specificity for two enzymes that function in consecutive reactions. ASAT2 and ASAT3, which are the second and third enzymes of the cultivated tomato F-type acylsucrose biosynthetic pathway, evolved new substrate specificities in the *S. habrochaites/S. pennellii* tomato subgroup. This allows the new Sp-ASAT3-P product to become a substrate for Sp-ASAT2-P, leading to reversal of their enzymatic order and pyranose ring P-type triacylsucrose production (Fig. 3c).

Discovery of the enzymes of P-type acylsucrose biosynthesis was facilitated by LC/ToF-MS screening of BILs, each of which contains multiple regions of the *S. pennellii* LA0716 genome in an *S. lycopersicum* genetic background[35]. The P-type acylsucrose producer (BIL6180) has three *S. pennellii* chromosomal regions, including those containing *Sp-ASAT2* and *Sp-ASAT3* (Fig. 1c). Evidence that these acylsucrose acyltransferases are sufficient to account for the P-type acylsucrose phenotype came from *S. lycopersicum* M82 transgenic lines expressing both *Sp-ASAT2* and *Sp-ASAT3* in type I/IV trichomes. These accumulated both F- and P-type acylsucroses (Fig. 2), as predicted based on the phenotype of a BIL6180 × wild-type M82 heterozygote (Supplementary

Fig. 2). Results of in vitro enzyme analysis confirmed that these enzymes produce P-type products (Fig. 3). Sp-ASAT3 uses monoacylsucrose S1:5 (iC5$^{R4}$) as an acyl acceptor substrate—as previously reported for Sl-ASAT2—to produce a diacylsucrose product S2:10 (iC5$^{R2}$, iC5$^{R4}$) that is distinct from any in the previously identified F-type acylsucrose pathway (Fig. 3c). Sp-ASAT2 next converts this product into P-type triacylsucroses. The combination of in vivo and in vitro results demonstrate that the 'flipped pathway' is at the center of divergence of the F- and P-type acylsucrose pathways (Fig. 3c).

Comparative biochemical analysis revealed a relatively small number of ASAT2 and ASAT3 amino acid differences that mediate the change in pathway order. ASAT2 neofunctionalization from F- to P-type enzymatic activity is associated with three amino acid substitutions (Fig. 4). *S. pennellii* and *S. habrochaites* accessions contain the ASAT2 change C304G, which is sufficient to promote ASAT2 P-type enzyme activity. In contrast, the phylogenetically more restricted Gln$^{135}$Tyr$^{136}$ to His$^{135}$Cys$^{136}$ change (Supplementary Figs. 11 and 14) abolishes the ASAT2 F-type activity in *S. pennellii* accessions from southern Peru (including LA0716, the *S. pennellii* parent of the IL and BIL). The differences between extant P- and F-type ASAT3 isoforms are mediated by changes in multiple amino acids, a theme common in enzyme evolution[42]. At least three residue substitutions—Y161H, C162S and T289V—are needed for the modern F-type Sl-ASAT3 to acquire P-type enzyme activity (Fig. 5b). Interestingly, comparison of all amino acids involved in flipping the pathway modeled the Sl-ASAT2 C304 and Sl-ASAT3 T289 to be at the same position in the predicted 3-D structures, near the acyl-CoA site (Supplementary Fig. 10d). These results suggest this position in the binding site as a candidate for future efforts to engineer BAHD acyltransferases for synthetic biology, and for understanding the evolution of form and function of related enzymes.

These results add to past work documenting how inter- and intra-specific variation in ASAT acyl donor promiscuity influence diversification of Solanum trichome acylsucrose products. For example, F-type ASAT2 enzymes from different tomato species specifically vary in preference for iC5-CoA as substrate[22], without affecting aiC5-CoA or nC12-CoA, thus influencing acyl chain types at the sucrose R₃ position[28]. In addition, all *S. habrochaites* F-type ASAT3 isoforms tested utilized iC5-CoA, but only a subset also accepted the long chain nC12-CoA donor[27]. Taken together with the results described above, we conclude that ASAT2 and ASAT3 acyl donor and acceptor promiscuity played roles in the evolution of these BAHD acyltransferases and diversification of trichome metabolite accumulation.

Despite the involvement of two enzymes, the divergence of F- and P-type acylsucrose appears to have occurred relatively recently—over the past 2.5 million years—which was a period of rapid diversification within the Solanum tomato clade[43,44]. We hypothesize that this innovation occurred following divergence leading to the *S. habrochaites/S. pennellii* group and away from the lineage containing the cultivated tomato *S. lycopersicum* and its closer relatives (shown by a red line in Fig. 7). We base this on two lines of evidence. First, ASAT2 and ASAT3 from the group of closer relatives (*S. lycopersicum* through *S. chilense*) exhibit F-type activities, whereas a mix of F- and P- type activities is observed for ASAT2 (Fig. 4) and ASAT3[27] from *S. habrochaites*. This suggests that the F-acylsucrose biosynthetic network is ancestral in the Solanum tomato clade. Second, based on extant *ASAT2* (Supplementary Fig. 11) and *ASAT3* (Supplementary Fig. 12) coding sequence phylogenetic relationships, only residues associated with F-type activities appear to have existed in the ancestor of wild tomatoes. In contrast, the gene phylogenies suggest that P-type residues emerged in an ancestor of *S. habrochaites* and *S. pennellii*. In addition, characterization of additional *ASAT3*

sequences from wild tomato species other than *S. habrochaites* reinforce the hypothesis that *ASAT3* duplication occurred after the last common ancestor of the extant tomato group and before *S. habrochaites* and *S. pennellii* speciation[27] (red line in Fig. 7).

While we have insufficient resolution to reconstruct the order of events leading to evolution of the P-type acylsucrose pathway, our results led to a general model for P-type pathway evolution, where promiscuity features prominently. For example, the C304G mutation is associated with P-type activity: mutagenesis of Sl-ASAT2 causes acquisition of P-type activity, while retaining the original F-type activity (Fig. 4b and Supplementary Fig. 6). This is not a merely a laboratory anomaly as there are ASAT2 natural variants that have both F- and P-type activities. For example, LA1777-ASAT2 cloned from *S. habrochaites* acylates both S1:5 (iC5$^{R4}$) and S2:10 (aiC5$^{R3}$, iC5$^{R4}$) (Fig. 4a). Similarly, substitution of Sl-ASAT3 with three amino acids associated with ASAT3-P activity (Y161H, C162S, T289V), led to S1:5 (iC5$^{R4}$) acylation on the pyranose ring, while retaining F-type activity (Supplementary Fig. 6). In fact, single substitution of each of these amino acids led to low detectable amounts of Sp-ASAT3 activity (Supplementary Fig. 9a, compare top three chromatograms with the bottom one). The promiscuity of such single or multiple mutants could have potentiated evolution of the ancestral enzyme, leading to production of small amounts of P-type diacylsucroses. In this scenario, accumulation of other mutations to the potentiated enzymes would eventually lead to improved catalytic activity, and a shift to the "purer" P-type activity seen in *S. pennellii* and *S. habrochaites*. ASAT2-P refinement by losing F-type activity presumably involved changes at positions 135 and 136 of Sl-ASAT2: Gln$^{135}$ to (His or Pro)$^{135}$ and Tyr$^{136}$ to Cys$^{136}$, based on extant sequences shown in Supplementary Fig. 11.

The results of this study leave interesting questions unanswered. For example, when did acylsugar degrading acylhydrolases[30] evolve and did these enzymes influence emergence or refinement of P-type enzymes? Did emergence of the ASAT3-P activity, which acylates pyranose at the same R$_2$ position as does the F-type pathway ASAT4 acetyltransferase, contribute to the ASAT4 gene inactivation events in ancestors of northern Ecuadoran *S. habrochaites* accessions[29] and *S. pennellii* LA0716[26]? Do differences in sucrose acylation patterns influence accumulation of acylglucoses in southern vs. northern *S. pennellii* accessions[45]? Thus, this work on variation in two acyltransferases opens up opportunities to study evolution of epistatic interactions between three or more enzymes in tomato and more broadly in the Solanaceae.

## Methods

**Plant growth and trichome acylsugar extraction.** Tomato wild species seeds were obtained from the C.M. Rick Tomato Genetic Resource Center (http://tgrc/ ucdvis.edu) and *S. pennellii* BIL seeds were from Dr. Dani Zamir (Hebrew University of Jerusalem). The tomato seeds were germinated on damp filter paper for one week and then transferred to peat pots. The plants were maintained in a growth chamber for another two to three weeks, with the growth condition as follows: 28 °C for 16 h in the light (300 µE m$^{-2}$ s$^{-1}$) and 20 °C for 8 h in the dark. Trichome metabolites were extracted by submerging the youngest fully developed leaf for 2 min with gentle agitation in 1 mL of extraction solution, which contained acetonitrile/isopropanol/water (3:3:2) with 0.1% formic acid and 10 µM propyl-4-hydroxybenzoate as internal standard. Dry leaf weight was measured after the extracted tissue was dried in an oven at 60 °C. We consider each individual plant as a biological replicate. At least three plants from the same line were used for the trichome acylsugar extraction and LC/MS analysis.

**Tomato transformation.** To express *Sp-ASAT2* and *Sp-ASAT3* in *S. lycopersicum* M82 trichomes, four fragments—a 1.6 kb region upstream of *Sl-ASAT2*; the *Sp-ASAT2* gene with a 1 kb region downstream; 1.7 kb region upstream of *Sl-ASAT3*, and the *Sp-ASAT3* gene—were amplified and cloned into the vector pENTR/D-TOPO (Invitrogen) using the Golden Gate Assembly method[46]. The gene cassette in pENTR/D-TOPO was recombined into the GATEWAY vector pK7WG[47] and used for transformation of M82 plants. To suppress *Sp-ASAT3* expression in

BIL6180 using RNAi, a fragment of *Sp-ASAT3* was amplified from *S. pennellii* LA0716 cDNA and cloned into pENTR/D-TOPO followed by recombination into the GATEWAY binary vector pHELLSGATE12[48] and the resultant plasmid was used to transform BIL6180 plants. Tomato transformation was performed using *A. tumefaciens* strain AGL0[49].

Trichome metabolite extractions were performed with the independent T0 primary transformants and T1 plants resulted from self-crossing of T0 lines. The primers used for genotyping of the T1 lines are described in Supplementary Table 1.

**Analysis of *Sp-ASAT3* and *Sp-ASAT2* gene expressions.** To analyze the mRNAs from *Sp-ASAT2* or *Sp-ASAT3* in T1 RNAi suppression lines targeting *Sp-ASAT3* in BIL6180, the leaves with the same developmental stages as those used for acylsugar extractions were collected. RNA extraction was performed using the Plant RNeasy kit (QIAGEN), and treated with DNase I. The first-strand cDNA synthesis by Superscript II (Invitrogen) used total RNA as templates. Five plants from each independent T1 lines were used as biological replicates. The Quantitative real-time PCR was performed to analyze the gene expression using the elongation factor gene (Solyc06g005060) as an internal control. Primers are listed in Supplementary Table 1. QuantStudio 7 Flex Real-Time PCR System with Fast SYBR Green Master Mix (Applied Biosystems) was used for the analysis. The relative quantification method ($2^{-\Delta\Delta Ct}$) was employed to evaluate the relative transcripts levels.

**LC/MS analysis of acylsugars.** Both trichome acylsugar extracts and enzyme assay samples were analyzed using a Waters Acquity UPLC system coupled to a Waters Xevo G2-S QToF LC-MS. For all sample injections, ten microliters were used for reverse-phase separation in a fused core Ascentis Express C18 column (2.1 mm × 10 cm, 2.7 µm particle size; Sigma-Aldrich) with the column temperature of 40 °C. The LC starting conditions were 95% solvent A (0.15% formic acid in water) and 5% solvent B (acetonitrile) with a flow rate of 0.3 mL/min. The 7 min and 14 min elution gradients – used for separating plant trichome acylsugars and enzyme assay samples—were as following steps. For the 7 min elution gradient: ramp to 40% B at 1 min, then to 100% B at 5 min, hold at 100% B to 6 min, return to 95% A at 6.01 min and hold until 7 min. For the 14 min elution gradient: ramp to 35% B at 1 min, then to 85% B at 12 min, ramp to 100% B at 12.01 min, hold to 13 min, return to 5% B at 13.01 min and hold until 14 min. For the MS settings: 2.14 kV capillary voltage, 90 °C source temperature, 350 °C desolvation temperature, 600 liters h$^{-1}$ desolvation nitrogen gas flow rate, 10 V cone voltage, and 50–1500 *m/z* mass range. Three separate acquisition functions were set up to generate spectra at different collision energies (5, 25, and 60 eV).

A Shimadzu LC-20AD HPLC system connected to a Waters LCT Premier ToF-MS was used for screening trichome acylsugars of the BILs. Ten microliters trichome metabolites samples were used for reverse-phase separation in a fused core Ascentis Express C18 column (2.1 mm × 10 cm, 2.7 µm particle size; Sigma-Aldrich) with the column temperature of 40 °C. The LC starting conditions were 90% solvent A (0.15% formic acid in water) and 10% solvent B (acetonitrile) with a flow rate of 0.4 mL/min. The 7-min elution gradient used to separate the trichome metabolites: ramp to 40% B at 1 min, then to 100% B at 5 min, hold at 100% B to 6 min, return to 90% A at 6.01 min and hold until 7 min. The 40-min elution gradient used to separate the BIL6180 acylsugars in Supplementary Fig. 1 employed the following linear steps: starting with 95% solvent A and 5% solvent B, ramp from 5% B to 10% B at 3 min, then to 34% B at 5 min, ramp to 35% B at 20 min, then to 58% B at 21 min, ramp to 59% B at 36 min, then to 100% B at 37 min and hold to 38 min, return to 5% B at 39 min and hold until 40 min. For the MS settings: 2.5 kV capillary voltage, 100 °C source temperature, 350 °C desolvation temperature, 350 liters h$^{-1}$ desolvation nitrogen gas flow rate, 10 V cone voltage, and 50–1500 *m/z* mass range. Three separate acquisition functions were set up to generate spectra at different collision energies (10, 40, and 80 eV).

Acylsugar annotations were inferred using collision-induced dissociation MS[25,28]. The carboxylate fragment ion masses generated by collision-induced dissociation MS in negative ion mode provided information about the number and length of acyl chains. Cleavage of the glycosidic linkage in positive ion mode by collision-induced dissociation MS yields information about acylation patterns of the pyranose and furanose ring of sucrose. Leaf acylsugar abundances were analyzed by the Waters QuanLynx tool to integrate extracted ion peak areas relative to the internal standard peak area, using the data generated in the negative-ion mode at the lowest collision energy (5 eV).

**Purification of acylsugars for NMR analysis.** Thirty BIL6180 plants were used to extract trichome acylsugars for NMR analysis. All plant leaves including the petiole tissues were harvested into a 5-liter beaker that contained 2 L of 100% methanol. After gentle agitation for 2 min, the solvents were transferred into two 1-liter glass bottles and the extraction solvent dried using a rotary evaporator under vacuum and then dissolved in 3 mL of 80% acetonitrile with sonication for 10 min. The concentrated solutions were centrifuged at 5000×g for 10 min at 25 °C to remove any insoluble debris and 200 µL aliquots were transferred to HPLC vials with glass inserts, and acylsugars were purified as described below.

The S2:10 (iC5, iC5) used for NMR analysis was made through sequential enzymatic reactions. In brief, the first enzymatic step was performed by incubating

100 µM iC5-CoA, 1 mM sucrose, and Sl-ASAT1 recombinant enzyme in 60 mL of 50 mM ammonium acetate (pH 6.0) buffer at 30 °C for 1 h, followed by heat inactivation of the enzyme at 65 °C for 10 min. The recombinant enzyme Sp-ASAT3 and 100 µM of iC5-CoA were then added to the solution for a 1 h incubation at 30 °C. The reactions were stopped by adding 120 mL acetonitrile/isopropanol/formic acid (1:1:0.001), and dried using a vacuum concentrator (SpeedVac[TM]) from Thermo Scientific followed by resuspension in 2 mL of water with ultrasonication for 10 min. Following centrifugation at 3000 g for 10 min at 25 °C, the supernatant was aliquoted into ten glass inserts in HPLC vials.

The acylsugars were purified using a Waters 2795 HPLC system coupled with an LKB BROMMA 221 Superrac fraction collector, using a 200 µL injection volume. A Thermo Scientific Acclaim 120 C18 HPLC column (4.6 × 150 mm, 5 µm particle size) was used for all compound separations with the column temperature set at 40 °C. Solvent A (0.15% formic acid in water) and solvent B (acetonitrile) were used as the LC mobile phase with the flow rate of 1.5 mL/min. The gradient elution methods used to separate the acylsugar compounds are as follows. For separating S3:15 (5,5,5) in BIL6180: 1% B at 0 min and hold to 2 min, ramp to 10% B at 3 min, then to 36% B at 5 min and hold to 33 min, ramp to 100% B at 34 min and hold to 38 min, return to 1% B at 39 min and hold to 40 min. S3:22 (5,5,12) from BIL6180 was purified as follows: 1% B at 0 min and hold to 2 min, ramp to 10% B at 3 min and then to 58% B at 5 min, ramp to 60% B at 33 min and then to 100% B at 34 min, hold 100% B to 38 min, then return to 1% B at 39 min and hold to 40 min. For separating the enzymatically produced S2:10 (iC5,iC5): 1% B at 0 min and hold to 2 min, ramp to 10% B at 3 min and then 14% B at 5 min, ramp to 15% B at 25 min and then to 100% B at 26 min, hold 100% B to 28 min, then return to 1% B at 29 min and hold to 30 min.

The eluted fractions were collected at a rate of 1 mL per minute. The fractions containing the desired compounds were verified by LC/MS, pooled and dried under vacuum. NMR spectra were recorded in solutions in Shigemi solvent-matched tubes (CDCl$_3$) for each purified compound using a Bruker Avance 900 NMR spectrometer equipped with a TCI triple-resonance inverse detection cryoprobe at the Michigan State University Max T. Rogers NMR facility. The $^1$H spectra were recorded at 899.27 MHz, and $^{13}$C NMR spectra were recorded at 226.14 MHz.

The NMR spectra data for S2:10 is as follows: $^1$H NMR: 5.66 (br., 1 H), 4.87 (br, m, 1 H), 4.86 (br, m, 1 H), 4.33 (t, $J = 8.3$ Hz, 1 H), 4.22 (d, $J = 7.5$ Hz, 1 H), 4.07 (t, $J = 9.9$ Hz, 1 H), 4.01 (br., 1 H), 3.82 (br, $J = 8.1$ Hz, 1 H), 3.72, 3.89 (m, $J = 12.6$ Hz, 2 H), 3.69 (m, $J = 13.2$ Hz, 2 H), 3.55, 3.63 (d, $J = 11.5$ Hz, 2 H), 2.28 (m, 2 H), 2.28 (m, 2 H), 2.12 (m, 1 H), 2.12 (m, 1 H), 0.98 (br, 6 H), 0.98 (br, 6 H); $^{13}$C NMR: 174.10, 173.10, 104.3, 89.03, 81.77, 78.66, 73.28, 72.43, 72.30, 70.36, 69.51, 64.32, 61.21, 59.61, 43.2, 42.27, 25.37, 25.37, 22.14, 22.14; HRMS ($m/z$, ESI-): calcd. for [M + formate]$^- = C_{23}H_{39}O_{15}^- = 555.2294$; found 555.2273.

The NMR spectra data for S3:15 is as follows: $^1$H NMR: 5.73 (m, $J = 3.6$ Hz, 1 H), 5.52 (m, $J = 9.8$ Hz, 1 H), 4.91 (m, $J = 9.9$ Hz, 1 H), 4.85 (m, $J = 10.2$ Hz, 1 H), 4.32 (t, $J = 8.3$ Hz, 1 H), 4.26 (d, $J = 8.3$ Hz, 1 H), 4.25 (m, 1 H), 3.77 (m, 1 H), 3.75, 3.86 (d, $J = 11.5$ Hz, 2 H), 3.64 (m, $J = 11.5$ Hz, 2 H), 3.49, 3.63 (m, 2 H), 2.39c (m, $J = 6.9$ Hz, 1 H), 2.37 (m, $J = 6.9$ Hz, 1 H), 2.30 (m, $J = 6.9$ Hz, 1 H), 2.22 (d, $J = 7.0$ Hz, 2 H), 2.20 (d, $J = 7.0$ Hz, 2 H), 2.12 (d, $J = 7.0$ Hz, 2 H), 2.05 (m, 1 H), 2.05 (m, 1 H), 2.00 (m, 1 H), 1.45, 1.67 (m, 2 H), 1.43, 1.63 (m, $J = 7.0$ Hz, 2 H), 1.39, 1.61 (m, 2 H), 1.12 (d, $J = 6.9$ Hz, 3 H), 1.09 (d, $J = 6.9$ Hz, 3 H), 1.06 (d, $J = 6.9$ Hz, 3 H), 0.94 (d, $J = 6.7$ Hz, 6 H), 0.94 (d, $J = 6.7$ Hz, 6 H), 0.93 (m, 3 H), 0.91 (m, 3 H), 0.91 (d, $J = 6.7$ Hz, 6 H), 0.90 (m, 3 H); $^{13}$C NMR: 176.98, 176.07, 175.90, 173.08, 172.38, 172.32, 104.90, 89.04, 81.64, 77.16, 73.02, 71.30, 70.61, 69.16, 68.55, 63.90, 61.48, 60.62, 42.95, 42.86, 42.56, 40.74, 40.65, 40.23, 26.33, 26.19, 26.19, 25.18, 25.13, 25.13, 22.66, 22.26, 21.81, 16.23, 16.20, 16.04, 11.47, 11.42, 11.26; HRMS ($m/z$, ESI-): calcd. for [M + formate]$^- = C_{28}H_{47}O_{16}^- = 639.2870$; found 639.2870.

The NMR spectra data for S3:22[a] is as follows: $^1$H NMR: 5.73 (m, $J = 3.2$ Hz, 1 H), 5.55 (t, $J = 9.9$ Hz, 1 H), 4.94 (m, $J = 10.1$ Hz, 1 H), 4.86 (dd, $J = 3.4$, 10.2 Hz, 1 H), 4.30 (t, $J = 8.3$ Hz, 1 H), 4.24 (d, $J = 7.9$ Hz, 1 H), 4.20 (m, 1 H), 3.81 (m, $J = 7.4$ Hz, 1 H), 3.74, 3.89 (d, $J = 13.2$ Hz, 2 H), 3.64 (m, $J = 12.5$ Hz, 2 H), 3.53, 3.61 (d, $J = 11.9$ Hz, 2 H), 2.40 (m, $J = 7.3$ Hz, 1 H), 2.37 (m, $J = 7.3$ Hz, 1 H), 2.22 (t, $J = 7.1$ Hz, 2 H), 1.54 (br, 2 H), 1.47, 1.61 (m, 2 H), 1.46, 1.69 (m, 2 H), 1.30 (m, 2 H), 1.25–1.27 (br, 14 H), 1.14 (d, $J = 6.7$ Hz, 3 H), 1.11 (d, $J = 6.9$ Hz, 3 H), 0.91 (m, 3 H), 0.89 (m, 3 H), 0.88 (m, 3 H); $^{13}$C NMR: 176.82, 176.14, 172.87, 88.80, 70.90, 68.60, 67.90, 40.95, 40.05, 33.75, 29.08–31.65, 26.53, 24.50, 22.21, 16.30, 15.64, 13.82, 11.20; HRMS ($m/z$, ESI-): calcd. for [M + formate]$^- = C_{35}H_{61}O_{16}^- = 737.3965$; found 737.3952.

The NMR spectra data for S3:22[b] is as follows: $^1$H NMR: 5.71 (d, $J = 3.05$ Hz, 1 H), 4.87 (dd, $J = 2.72$, 10.2 Hz, 1 H), 2.40 (m, $J = 6.8$ Hz, 1 H), 1.13 (d, $J = 6.7$ Hz, 3 H), 1.44, 1.61 (m, $J = 7.2$ Hz, 2 H), 0.87 (m, 3 H), 5.52 (m, $J = 9.7$ Hz, 1 H), 2.21 (m, $J = 7.1$ Hz, 2 H), 1.53 (br, 2 H), 1.27–1.29 (br, 14 H), 1.29 (m, $J = 6.8$ Hz, 2 H), 0.87 (m, 3 H), 4.91 (m, $J = 9.9$ Hz, 1 H), 2.18 (d, $J = 7.0$ Hz, 2 H), 2.05 (m, 1 H), 0.93 (m, $J = 7.0$ Hz, 6 H), 4.23 (m, 1 H), 3.60, 3.64 (m, $J = 12.0$ Hz, 2 H), 3.49, 3.60 (d, $J = 8.7$ Hz, 2 H), 4.26 (m, 1 H), 4.26 (m, 1 H), 3.77 (m, 1 H), 3.75, 3.87 (d, $J = 11.8$ Hz, 2 H); $^{13}$C NMR: 176.92, 173.03, 172.25, 104.5, 88.9, 81.58, 77.82, 73.27, 71.7, 70.78, 69.16, 68.74, 64.4, 61.7, 60.8, 42.63, 40.30, 33.76, 29.29–32.05, 26.50, 25.09, 24.52, 22.35, 21.79, 15.75, 13.86, 11.13; HRMS ($m/z$, ESI-): calcd. for [M + formate]$^- = C_{35}H_{61}O_{16}^- = 737.3965$; found 737.3967.

### Site-Directed mutagenesis

A PCR-based site-directed mutagenesis method was used to introduce mutations to *ASATs* with the help of Q5 Site-Directed Mutagenesis Kit (NEB). The presence of mutations was validated by Sanger DNA sequencing of the pET28b plasmids with the genes. The web-based software NEBaseChanger (Version 1.2.2, NEB) was used to design primers for generating mutations, which are listed in Supplementary Table 1.

### Protein expression and enzyme assays

Restriction digest cloning was used to clone various *ASAT2* or *ASAT3* orthologs into the pET28b expression vector. The recombinant proteins were produced using *Escherichia coli* BL21 Rosetta cells (EMD Millipore) as the host, as described[22]. Nickel affinity chromatography was used to purify the soluble proteins[26].

Activity assays for ASAT2 and ASAT3 were performed by incubating purified recombinant enzymes in 30 µL of 50 mM ammonium acetate (pH 6.0) buffer with 100 µM acyl-CoAs. 50 µM acylsucrose acceptor—including purified S1:5 (iC5$^{R4}$), S2:10 (iC5$^{R2}$, iC5$^{R4}$), and S2:10 (iC5$^{R4}$, aiC5$^{R3}$)—were used to test F- and P-type activities of different ASAT2 and ASAT3 orthologs. The assays were kept at 30 °C for 30 min and the reactions terminated by addition of 60 µL of acetonitrile/isopropanol/formic acid (1:1:0.001) with 10 µM propyl-4-hydroxybenzoate as the internal standard. The reaction solutions were centrifuged at 13,000×g for 10 min and the supernatant transferred to glass inserts in HPLC vials for LC/MS analysis. All enzyme reactions were performed at least three times. The representative LC/MS chromatograms from one of the assays are shown in the figures. The acyl-CoA substrates—iC5-CoA and nC12-CoA—were purchased from Sigma-Aldrich. Chemical synthesis of aiC5-CoA was as described[27].

### ASAT orthologs amplification and phylogenetic analysis

*ASAT2* orthologs cloned from *S. lycopersicum* through *S. corneliomulleri* that were used to build the phylogenetic tree in Supplementary Fig. 11 are from a previous study[22]. We amplified additional *ASAT2* ortholog sequences from additional accessions of *S. habrochaites* and *S. pennellii* genomic DNA using the primers listed in Supplementary Table 1. In the previous report, multiple *ASAT3-F* and *ASAT3-P* type alleles were amplified from different accessions of *S. habrochaites* using the F- and P-type primer sets[27]. In the current work, the same primer sets were used to amplify F- and P-type *ASAT3* alleles in multiple wild tomato species. cDNA was used as templates to amplify *ASAT3* from *S. pimpinellifolium*, *S. galapagense*, *S. neorickii*, *S. arcanum*, *S. chilense*, *S. peruvianum*, and *S. corneliomulleri*. For *S. pennellii* accessions, genomic DNA was used to amplify *ASAT3*. The amplified products were cloned into pET28b using the restriction cloning method and the *ASAT2* or *ASAT3* sequences obtained by Sanger sequencing using plasmids extracted from individual clones. The sequences are in GeneBank with the accession numbers listed below.

For each *ASAT2* ortholog, the full coding sequences were used to build the phylogenetic tree. The regions introduced by F- or P- forward primers—24 bp in the 5′ end of the *ASAT3* sequences—were eliminated from the phylogenetic analysis to avoid hybrid coding sequences. The nucleotide sequences were aligned using the default MUSCLE algorithm with the MEGA7 software[50]. For both *ASAT2* and *ASAT3*, the Tamura 3-parameter Maximum Likelihood model was chosen for phylogenetic tree construction from 24 different nucleotide substitution models based on the lowest Bayesian Information Criterion. The bootstrap values were obtained after 1000 replicates. The phylogenetic trees in Fig. 4a, Supplementary Figs. 11 and 12 were all made using the above method.

### Homology structural modeling of Sl-ASAT2 and Sl-ASAT3

The Phyre2 web portal[51] was used to predict the tertiary structures of Sl-ASAT2 and Sl-ASAT3. The Open-Source PyMOL software (Version 1.8.X Schrödinger, LLC) was applied to overlay the Sl-ASAT2 and Sl-ASAT3 modeled structures with the trichothecene 3-*O*-acetyltransferase structure (PDB code 3B2S)[38].

### Data availability

The data that support the findings of this study are available from the corresponding author upon reasonable request. The NMR raw data for the four purified compounds S2:10, S3:15, S3:22[a], and S3:22[b] are deposited in the public data repository Open Science Framework under the link https://osf.io/exfmr/. Sequence data that support the findings of this study are available as GenBank/EMBL data libraries under these accession numbers. ASAT2: Sl-ASAT2 (KT359567), Sp-ASAT2 (KT359557), LA1578-ASAT2 (KT359558), LA1278-ASAT2 (KT359559), LA2133-ASAT2 (KT359560), LA1777-ASAT2 (KT359561), LA1401-ASAT2 (KT359562), LA0107-ASAT2 (KT359563), LA1969-ASAT2 (KT359564), LA2172-ASAT2 (KT359565), LA1718-ASAT2 (KT359566), Sl-ASAT3 (KM516150), LA1718-ShASAT2_2 (KY962559), LA1737-ShASAT2_1 (KY962560), LA1737-ShASAT2_2 (KY962561), LA1737-ShASAT2_3 (KY962562), LA1772-ShASAT2_1 (KY962563), LA1772-ShASAT2_2 (KY962564), LA1978-ShASAT2_1 (KY962565), LA1978-ShASAT2_2 (KY962566), LA1978-ShASAT2_3 (KY962567), LA2098-ShASAT2_1 (KY962568), LA2098-ShASAT2_2 (KY962569), LA2156-ShASAT2_1 (KY962570), LA2156-ShASAT2_2 (KY962571), LA2156-ShASAT2_3 (KY962572), LA2196-ShASAT2_1 (KY962573), LA2196-ShASAT2_2 (KY962574), LA2650-ShASAT2_1 (KY962575), LA2650-ShASAT2_2 (KY962576), LA2722-ShASAT2_1 (KY962577), LA2722-ShASAT2_2 (KY962578), LA2861-ShASAT2_1 (KY962579), LA2861-ShASAT2_2 (KY962580), LA2861-ShASAT2_3 (KY962581), LA2975-ShASAT2_1 (KY962582), LA2975-ShASAT2_2 (KY962583), LA2975-ShASAT2_3 (KY962584), LA1282-SpASAT2 (KY962585), LA1356-

SpASAT2 (KY962586), LA1367-SpASAT2 (KY962587), LA1376-SpASAT2 (KY962588), LA1649-SpASAT2 (KY962589), LA1911-SpASAT2 (KY962590), LA1926-SpASAT2 (KY962591), LA1946-SpASAT2 (KY962592), LA2560-SpASAT2_1 (KY962593), LA2560-SpASAT2_2 (KY962594), LA2963-SpASAT2 (KY962595).

ASAT3: Sl-ASAT3 (KM516150), Sp-ASAT3 (KM516151), LA1777-ShASAT3-F (KM516152), LA1978-ShASAT3-F (KM516153), LA2156-ShASAT3-F (KM516154), LA2204-ASAT3-like1 (KM516155), LA2861-ShASAT3-F (KM516156), LA2861-ShASAT3-P (KM516157), LA2156-ASAT3-like (KM516158), LA1777-ShASAT3-P (KM516159), LA1731-ShASAT3-P (KM516160), LA1731-ShASAT3-F (KM516161), LA2722-ShASAT3-P (KM516162), LA2650-ShASAT3-P (KM516163), LA2574-ShASAT3-P (KM516164), LA2204-ASAT3-like2 (KM516165), LA1772-ShASAT3-P (KM516166), LA2098-ShASAT3-P (KM524335), LA1926-SpASAT3 (KY962596), LA2560-SpASAT3 (KY962597), LA1946-SpASAT3 (KY962598), LA1941-SpASAT3 (KY962599), LA1911-SpASAT3 (KY962600), LA1649-SpASAT3 (KY962601), LA1376-SpASAT3_2 (KY962602), LA1367-SpASAT3_2 (KY962603), LA1367-SpASAT3_1 (KY962604), LA1356-SpASAT3_2 (KY962605), LA1356-SpASAT3_1 (KY962606), LA1282-SpASAT3_3 (KY962607), LA1282-SpASAT3_2 (KY962608), LA1282-SpASAT3_1 (KY962609), LA2963-SpASAT3 (KY962610), LA1376-SpASAT3_1 (KY962611), LA1578-ASAT3_1 (KY962612), LA1578-ASAT3_2 (KY962613), LA2133-ASAT3 (KY962614), LA2172-ASAT3_1 (KY962615), LA2172-ASAT3_2 (KY962616), LA1364-ASAT3 (KY962617), LA1401-ASAT3 (KY962618), LA1969-ASAT3 (KY962619), LA1278-ASAT3_1 (KY962620), LA1278-ASAT3_2 (KY962621), LA0107-ASAT3_2 (KY962622), LA0107-ASAT3_1 (KY962623), LA0107-ASAT3_3 (KY962624).

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

## Acknowledgements

We thank the C.M. Rick Tomato Genetics Resource Center (University of California Davis, CA USA) for providing wild tomato species seeds, Zamir lab in Hebrew University of Jerusalem for providing tomato BILs seeds. We acknowledge members of the Solanum Trichome Project for their contributions to this work: especially Kathleen Imre for her help with tomato transformations; Anthony Schilmiller for providing plasmid constructs; Bryan Leong, Gaurav Moghe, and Daniel Lybrand for their helpful comments on the manuscript. We acknowledge the MSU Center for Advanced Microscopy and RTSF Mass Spectrometry and Metabolomics Core Facilities for their support with LC/MS analysis. This work was funded by National Science Foundation grants IOS-1025636 and IOS-PGRP-1546617 to A.D.J. and R.L.L. Abigail Miller was supported by NSF REU grant DBI-1358474 in the summer of 2014 and an American Society of Plant Biologists Summer Undergraduate Research Award during the summer of 2015. A.D.J. acknowledges support from the USDA National Institute of Food and Agriculture, Hatch project MICL-02143.

## Author contributions

P.F. and R.L.L. designed research; P.F. and A.M.M., performed experiments; X.L. performed NMR analyses; P.F., A.M.M., X.L., A.D.J., and R.L.L. analyzed data; P.F. and R.L.L. wrote the manuscript. All authors read and edited the manuscript.

## Additional information

**Competing interests:** The authors declare no competing financial interests.

