## [Peer Review File · Nature Communications]

Reviewers' comments:

Reviewer #1 (Remarks to the Author):

The major claims of the manuscript entitled 'Metabolic innovation through evolutionary pathway engineering in glandular trichomes of tomato' are that diversity of acylated sugars in tomato have arisen through small amino acid changes in key acyltransferases of the BAHD superfamily. Furthermore, the modification of two key acyltransferases can change the accumulation profile of acylsucroses in wild tomato plants. The authors used a series of backcrossed inbred lines followed by targeted metabolite analysis to identify Sp-ASAT2 and Sp-ASAT3 as the acyltransferases responsible for the F to P type switching.

The work presented in this manuscript is of the highest quality as well as the writing of the report itself. For the most part, the figures are well done and the legends are descriptive. However, I have several minor concerns:

Figure 1. The Y axis for the chromatograms in section B are only labeled in terms of %. While the figure also contains the TIC intensity other readers may not be able to interpret this. Even a text of % of total ions would be more helpful.

Figure 3a There is no label for the Y-axis at all here. In addition, there is no intensity of the TIC for either A or B in this figure. Why didn't the authors report the specific activities of their enzymes in the in vitro enzyme assays? They reported purifying the proteins (although no gels of the purification could be found in the sI) and performing the assays. In the absence of kinetics, specific activities under their conditions is needed.

In testing their 'flipped pathway' hypothesis, the authors utilized RNAi lines of Sp-ASAT3 and observed a dramatic reduction in total acylsucroses. The authors use this as evidence to reinforce their hypothesis. However, there are several critical controls that are not reported or in the supplemental material. First, the authors did not measure Sp-ASAT3 transcripts in their RNAi lines to show a reduction in overall transcript levels. This should also be performed with Sp-ASAT2 gene transcript levels to show that the RNAi is not targeting both genes. Furthermore, RNAi experiments of Sp-ASAT2 lines should show an accumulation of S2:10 (iC5R2,iC5R4) diacylsucrose if this flipped hypothesis is correct. Why did the authors not choose to perform this experiment?

Figure 4a shows a phylogram made using different ASAT2 orthologs. There is no description in the figure legend how this tree was made, what the numbers at the nodes represent (assumed to be bootstraps, or what the scale bar represents). The tree looks to be rooted, but its root (Sotub04g011000) is not described. If the numbers at the nodes, are bootstraps, there is no mention how many replicates were performed. Indeed, bootstraps as low as 30 or 50 (below 80) need to be collapsed at that particular node as there is little confidence in their placement. The methods section only describes trees made using nucleotide sequence. Is this tree made using amino acid sequences? If so, what algorithms were used to generate it and perform the original alignment?

Reviewer #2 (Remarks to the Author):

The role of enzyme promiscuity is increasingly becoming recognized as an important driver of chemical diversity in plant natural product biosynthesis, and evolutionary processes are at the heart of these concepts. Unfortunately, many times plant biochemist may not be fully aware of the evolutionary implications of their studies, and likewise, many evolutionary biologists have had little training in biochemistry. Fan et al bridge these two disciplines and weave together an interesting story describing how acylsucrose structural diversity has evolved across different tomato lineages. Building on their previous identification of the key BAHD acyltransferases, they bring together an

impressive amount of work to build a strong case explaining how neofunctionalization of ASAT2 and ASAT3 underscore the chemical diversity observed in not only different varieties but also different geographical regions. Importantly, this study details the molecular basis and specific amino acids that underpin the differences in acylsucrose diversity, and how this pathway could have “flipped,” enabling a new metabolic innovation of P-type acylsucroses.

The authors have generated a solid amount of data to elucidate much of the structural basis for the enzymatic promiscuity, to the point where the authors show that they can swap the enzymatic function by introducing specific amino acid substitutions to both ASAT2 and ASAT3. This in itself is a feat that will be relevant to protein engineers given the broad interest of BAHD acyltransferases in many other natural product biosynthetic pathways. Moreover, Fan et al also build upon these efforts by heterologously expressing and silencing pathways, providing strong support that ASAT2 and ASAT3 neofunctionalized after the divergence of the *S. habrochaites*/*S. pennellii* clade from the rest of cultivated tomatoes. Our understanding of evolutionary processes is built on anecdotal and empirical studies, and this study provides a relevant case study in understanding how enzyme promiscuity may lead to a new biosynthetic pathway.

Minor issues:

The authors may want to consider changing the title, as it is a bit vague and potentially misleading. It is not clear what the term ‘evolutionary pathway engineering’ is supposed to mean (one thinks of adaptive evolutionary experiments) and deviates from the core findings of the study which are that enzyme promiscuity enabled a novel metabolic pathway to arise.

There should be more discussion as to what the authors believe might be the selective pressures driving these metabolic innovations or losses. Specifically, it would be appreciated if there could be more text devoted to commenting on what is known (or not known) about the physiological role of P-type vs F-type acylsugars for the plant.

Another interesting finding that would be worth fleshing out a bit more is the result that LA1364 and LA1278 ASAT3 sequences seem to have undergone convergent evolution to also have P-type activity via His161 and Ser162. What are the chances that these residues have convergently evolved with the *S. pennellii* and *S. habrochaites* orthologs to carry out P-type activity? Is there a reason to think that P-type acyl sugars may confer some selective advantage?

Reviewer #3 (Remarks to the Author):

The report by Fan and colleagues uses a combination of genetic tools with site directed mutagenesis and with enzyme analyses to identify the detailed biochemical and potential evolutionary mechanisms responsible for the formation of acylated sugars in tomato glandular trichomes. This study follows a series of interesting and original studies to characterize the acyltransferases responsible for various sucrose acylation patterns in surface glands of tomato and ancestors in relation to their evolutionary origins in various parts of South America. The study is well executed and the majority of the conclusions are justified by the results presented. There are a few questions critiques that hopefully can be addressed by the authors:

The genetic and metabolic evidence in Fig. 1, 2 and S2 provide convincing data to suggest that Sp-ASAT2 and Sp-ASAT3 are likely responsible for biosynthesis of P-type acylsucrose formation in *S. pennellii*.

- These suggestions are confirmed by genetic engineering of *S. lycopersicum* with Sp-ASAT2 and Sp-ASAT3 in trichomes and showing that P-type acylsucrose began to accumulate in transgenic lines.
- While the data presented in Fig 2, S2 and S3 help to suggest this, it is important for the authors to provide more quantitative information than ‘peak area/internal standard’?

There is no report on the variation observed: have the authors made biological replicates and technical replicates of this data? This should be part of the presentation.

- The authors might consider providing the reader an idea of how similar are the levels of P-acylsugars produced (low to high) in various transformation events compared to those found in *S. pennelli*? This reviewer expects the levels might be similar to those found in *S. pennelli*?

The biochemical assays conducted with recombinant Sp-ASAT2 and Sp-ASAT3 enzymes with data shown in Fig 3 provide strong evidence to show that the enzyme actions of Sp-ASAT2 and Sp-ASAT3 are reversed to those of *S. lycopersicum*.

- The figure legend in 3c is confusing to readers not familiar with these types of reactions: ["S" and "L" refer to short chain iC5-CoA or long chain nC12-CoA substrates, respectively]. The authors need to describe that enzyme assays use the acylated sugar acceptor and different CoA Esters.
- In this context, how do the authors explain the perceived low levels of nC12-ester being produced by this enzyme compared with iC5-ester?
- Is it possible that a different acyltransferase is involved in this reaction? The results presented in this figure are qualitative? What would the quantitative differences be?
- Given some of the promiscuity of some acyltransferases, is it not possible that a separate gene might be involved in generating longer chain esters? Perhaps the transgenic RNAi results back this possible role of Sp-ASAT3 and this should be stated more explicitly?
- Another way to study this would be to look at enzymes extracted directly from trichomes to see how actively acylation would take place with native enzymes when the nC12-CoA the acyl acceptor was supplied to enzyme assays. While the evidence produced would be circumstantial, it might corroborate the low activities observed with recombinant proteins. It is remarkable that these types of studies with enzyme extracts from the plant rarely take place any more.

The authors follow the recombinant enzyme studies with site specific mutagenesis studies to show that a few amino acids are likely to be responsible for the functional divergence of the F- and P-type acyl-sugars found in different tomato species found in different geographical areas of South America.

- While the data from mutagenesis studies provides evidence of the amino acid substitutions involved, the data is qualitative. It would be of interest to know more about the detailed kinetic changes that might be generated by these substitutions (K_m , V_{max} , K_{cat})

Other comments/suggestions

1. Please explain the meaning of the following statement: "Line 185-87 [Published work demonstrated that SI-ASAT2 acylates the product of SI-ASAT1 and Sp-ASAT1 — monoacylsucrose S1:5 (iC5R4) — at the R3 position to produce a diacylsucrose with both chains on the pyranose ring]. Do you mean [Published work demonstrated that SI-ASAT2 acylates monoacylsucrose S1:5 (iC5R4), the product of SI-ASAT1 or Sp-ASAT1 reactions at the R3 position to produce a diacylsucrose on the pyranose ring]?"
2. Line 191-92: Meaning of: "Purification and NMR analysis of the diacylsucrose S2:10 (iC5, iC5R4) produced by the combination of SI". Do you mean: "NMR analysis of the purified diacylsucrose S2:10 (iC5, iC5R4) produced by the combination of SI"
3. Line 238-240: Meaning: "In vitro mutagenesis of C304G in SI-ASAT2 led to the ability to use S2:10 (iC5R2, iC5R4) and produce the P-type triacylsucrose S3:22 (iC5R2, nC12R3, iC5R4) (Fig. 4b)." Is this better: "In vitro mutagenesis of C304G in SI-ASAT2 led to the ability to convert S2:10 (iC5R2, iC5R4) to the P-type triacylsucrose S3:22 (iC5R2, nC12R3, iC5R4) (Fig. 4b)."
4. Line 329: "holding a unique position to inform us regarding the transition between F- and P-type acylsucrose pathways."
5. Lines 364-66: Meaning of "Evidence that these acylsucrose acyltransferases are sufficient for the P-type acylsucrose phenotype came from *S. lycopersicum* M82 transgenic lines expressing both Sp-ASAT2 and Sp-ASAT3 in type I/IV trichomes." "Evidence that these acylsucrose acyltransferases are sufficient to account for the P-type acylsucrose phenotype came from *S.*

lycopersicum M82 transgenic lines expressing both Sp-ASAT2 and Sp-ASAT3 in type I/IV trichomes.'

Reviewer #1 (Remarks to the Author):

The major claims of the manuscript entitled 'Metabolic innovation through evolutionary pathway engineering in glandular trichomes of tomato' are that diversity of acylated sugars in tomato have arisen through small amino acid changes in key acyltransferases of the BAHD superfamily. Furthermore, the modification of two key acyltransferases can change the accumulation profile of acylsucroses in wild tomato plants. The authors used a series of backcrossed inbred lines followed by targeted metabolite analysis to identify Sp-ASAT2 and Sp-ASAT3 as the acyltransferases responsible for the F to P type switching.

The work presented in this manuscript is of the highest quality as well as the writing of the report itself. For the most part, the figures are well done and the legends are descriptive.

However, I have several minor concerns:

Figure 1. The Y axis for the chromatograms in section B are only labeled in terms of %. While the figure also contains the TIC intensity other readers may not be able to interpret this. Even a text of % of total ions would be more helpful.

The Y axis for the chromatograms in section B is relative abundance of the negative-ion mode LC/MS chromatogram peak intensity. For better comparison, the two chromatograms were scaled to the same TIC intensity, where the major peak (S4:17-F) in M82 has the maximum signal (100% scale). The label was added and the figure legend was adjusted in the revised manuscript.

Figure 3a There is no label for the Y-axis at all here. In addition, there is no intensity of the TIC for either A or B in this figure.

The label for Y-axis and the intensity of the TIC were added to Figure 3a.

Why didn't the authors report the specific activities of their enzymes in the in vitro enzyme assays? They reported purifying the proteins (although no gels of the purification could be found in the sI) and performing the assays. In the absence of kinetics, specific activities under their conditions is needed.

We agree that kinetic information is desirable, but are constrained by experimental considerations. We cannot provide specific activities of the enzymes due to the lack of ability to quantify the absolute concentrations of the remaining acylsugar substrates or new acylsugar products after the enzyme reactions. Even though we are not able to provide quantification data of the enzyme activities, the peak areas of acylsugar products are comparable across same sets of enzyme reactions given equal amount of substrates and enzymes under controlled reaction conditions. Furthermore, the main conclusions drawn from the results are largely based on qualitative analysis whether or not the enzymes or enzyme mutants can catalyze the given substrates.

We would like to measure the K_m of the acyl acceptor substrates of the enzymes, which can be calculated using the product peak areas. However, the acyl acceptor substrates [S1:5(iC5^{R4}), S2:10(aiC5^{R3}, iC5^{R4}), and S2:10(iC5²³, iC5^{R4})] are not commercially available. And the ones we purified from large scale enzyme reactions cannot be produced economically or concentrated enough to saturate the ASATs. Sl-ASAT1's acyl acceptor substrate (sucrose) had a K_m of 2.3 mM (Fan et al, PNAS, 2016). In that experiment, the highest sucrose concentration used to saturate Sl-ASAT1 was 20 mM. We have a reason to predict that the tomato ASATs share relatively high K_m for acyl acceptor substrates and the concentration as high as 200 mM might be needed for the kinetics analysis.

In testing their 'flipped pathway' hypothesis, the authors utilized RNAi lines of Sp-ASAT3 and observed a dramatic reduction in total acylsucroses. The authors use this as evidence to reinforce their hypothesis. However, there are several critical controls that are not reported or in the supplemental material. First, the authors did not measure Sp-ASAT3 transcripts in their RNAi lines to show a reduction in overall transcript levels. This should also be performed with Sp-ASAT2 gene transcript levels to show that the RNAi is not targeting both genes.

Based on this request, we characterized T1 lines from five independent T0 transgenic lines. Five PCR-confirmed transgenic plants from each T1 line were analyzed. The new results are shown in Supplementary figure 5 (b) and (c). The total acylsugar level reduction was observed in all five independent T1 lines, as well as the reduction of Sp-ASAT3 transcript levels. Analysis of Sp-ASAT2 transcript levels revealed that RNAi is not targeting both genes.

Furthermore, RNAi experiments of Sp-ASAT2 lines should show an accumulation of S2:10(iC5R2,iC5R4) diacylsucrose if this flipped hypothesis is correct. Why did the authors not choose to perform this experiment?

*During the course of the study we considered silencing Sp-ASAT2 in BIL6180 to ask whether S2:10 ($iC5^{R2}$, $iC5^{R4}$) accumulated but chose not to do this analysis. This is because our previous experiments silencing Sl-ASAT3 in *S. lycopersicum* M82 led to low amounts of S2:17 observed and no S2:10 (Schilmiller et al, Plant Cell, 2015).*

Figure 4a shows a phylogram made using different ASAT2 orthologs. There is no description in the figure legend how this tree was made, what the numbers at the nodes represent (assumed to be bootstraps, or what the scale bar represents. The tree looks to be rooted, but its root (Sotub04g011000) is not described. If the numbers at the nodes, are bootstraps, there is no mention how many replicates were performed. Indeed, bootstraps as low as 30 or 50 (below 80) need to be collapsed at that particular node as there is little confidence in their placement. The methods section only describes trees made using nucleotide sequence. Is this tree made using amino acid sequences? If so, what algorithms were used to generate it and perform the original alignment?

*The phylogenetic tree in Figure 4a was originally made using amino acid sequences. In the revised manuscript, we remade the tree using nucleotide sequences to be consistent with other phylograms in the manuscript. The ASAT2 homolog (Sotub04g011000) in *S. tuberosum* was used as the root. The nucleotide sequences were aligned using the default MUSCLE algorithm. The Tamura 3-parameter Maximum Likelihood model was chosen for phylogenetic tree construction from 24 different nucleotide substitution models based on the lowest Bayesian Information Criterion. The bootstrap values were obtained after 1000 replicates. The nodes with bootstrap values lower than 70 were collapsed.*

Reviewer #2 (Remarks to the Author):

The role of enzyme promiscuity is increasingly becoming recognized as an important driver of chemical diversity in plant natural product biosynthesis, and evolutionary processes are at the heart of these concepts. Unfortunately, many times plant biochemist may not be fully aware of the evolutionary implications of their studies, and likewise, many evolutionary biologists have had little training in biochemistry. Fan et al bridge these two disciplines and weave together an interesting story describing how acylsucrose structural diversity has evolved across different tomato lineages. Building on their previous identification of the key BAHD acyltransferases, they bring together an impressive amount of work to build a strong case explaining how neofunctionalization of ASAT2 and ASAT3 underscore the chemical diversity observed in not only different varieties but also different geographical regions. Importantly, this study details the molecular basis and specific amino acids that underpin the differences in acylsucrose diversity, and how this pathway could have “flipped,” enabling a new metabolic innovation of P-type acylsucroses.

The authors have generated a solid amount of data to elucidate much of the structural basis for the enzymatic promiscuity, to the point where the authors show that they can swap the enzymatic function by introducing specific amino acid substitutions to both ASAT2 and ASAT3. This in itself is a feat that will be relevant to protein engineers given the broad interest of BAHD acyltransferases in many other natural product biosynthetic pathways. Moreover, Fan et al also build upon these efforts by heterologously expressing and silencing pathways, providing strong support that ASAT2 and ASAT3 neofunctionalized after the divergence of the *S. habrochaites*/*S. pennellii* clade from the rest of cultivated tomatoes. Our understanding of evolutionary processes is built on anecdotal and empirical studies, and this study provides a relevant case study in understanding how enzyme promiscuity may lead to a new biosynthetic pathway.

Minor issues:

The authors may want to consider changing the title, as it is a bit vague and potentially misleading. It is not clear what the term ‘evolutionary pathway engineering’ is supposed to mean (one thinks of adaptive evolutionary experiments) and deviates from the core findings of the study which are that enzyme promiscuity enabled a novel metabolic pathway to arise.

Thank you for this perspective. We suggest a new title: “Evolution of a flipped biosynthetic pathway creates metabolic innovation in glandular trichomes of tomato through BAHD enzyme promiscuity.”

There should be more discussion as to what the authors believe might be the selective pressures driving these metabolic innovations or losses. Specifically, it would be appreciated if there could be more text devoted to commenting on what is known (or not known) about the physiological role of P-type vs F-type acylsugars for the plant.

Another interesting finding that would be worth fleshing out a bit more is the result that LA1364 and LA1278 ASAT3 sequences seem to have undergone convergent evolution to also have P-type activity via His161 and Ser162. What are the chances that these residues have convergently evolved with the *S. pennellii* and *S. habrochaites* orthologs to carry out P-type activity? Is there a reason to think that P-type acyl sugars may confer some selective advantage?

We agree that LA1363 and LA1278 potential represent convergent evolution. However, the lack of P-type ASAT2 in these accessions presumably mask their impact on triacylated product accumulation. Thus we prefer to avoid introducing a potentially confusing new line of reasoning. Do you agree?

*We have no information regarding differences in biotic stress sensitivities or tolerance associated with P- versus F- type acylsugars. However, we found that *S. pennellii* acylglucose biosynthesis is enabled by the ability of an invertase like enzyme to cleave P-type – and not F-type – acylsucroses. Thus, it appears that the flipped pathway potentiated production of acylglucoses. We are completing this study and plan to submit it for publication in the future.*

Reviewer #3 (Remarks to the Author):

The report by Fan and colleagues uses a combination of genetic tools with site directed mutagenesis and with enzyme analyses to identify the detailed biochemical and potential evolutionary mechanisms responsible for the formation of acylated sugars in tomato glandular trichomes. This study follows a series of interesting and original studies to characterize the acyltransferases responsible for various sucrose acylation patterns in surface glands of tomato and ancestors in relation to their evolutionary origins in various parts of South America. The study is well executed and the majority of the conclusions are justified by the results presented. There are a few questions critiques that hopefully can be addressed by the authors:

The genetic and metabolic evidence in Fig. 1, 2 and S2 provide convincing data to suggest that Sp-ASAT2 and Sp-ASAT3 are likely responsible for biosynthesis of P-type acylsucrose formation in *S. pennellii*.

- These suggestions are confirmed by genetic engineering of *S. lycopersicum* with Sp-ASAT2 and Sp-ASAT3 in trichomes and showing that P-type acylsucrose began to accumulate in transgenic lines.
- While the data presented in Fig 2, S2 and S3 help to suggest this, it is important for the authors to provide more quantitative information than ‘peak area/internal standard?’

There is no report on the variation observed: have the authors made biological replicates and technical replicates of this data? This should be part of the presentation.

In the revised manuscript, we normalized the peak area/internal standard using the weight of the dried leaves for better comparison among different T0 lines.

As we consider each plant as a biological sample, no biological replicate can be analyzed in T0 generation. Therefore, new characterizations were performed in the selfed T1 generation by germinating the seeds from four independent T0 lines. Five plants from each T1 lines carrying the transgene --identified by PCR genotyping -- were analyzed for acylsugars. The other transgenic lines using RNAi to suppress Sp-ASAT3 in BIL6180 were also further characterized in the T1 generation with biological replicates from each independent T1 lines. The new results are shown in Supplementary figure 3 (b) and 5 (b)(c).

- The authors might consider providing the reader an idea of how similar are the levels of P-acylsugars produced (low to high) in various transformation events compared to those found in *S.pennelli*? This reviewer expects the levels might be similar to those found in *S. penelli*?

*The design of this experiment was intended to show whether new P-acylsugars could be made in *S. lycopersicum* — a non P-acylsugars producing plant — after introducing two *S. pennellii* genes. We prefer not to focus on quantitative differences for two reasons. First, the expression levels of the two Sp-ASATs are not*

comparable to *S. pennellii* in the transgenic lines. The gene promoters used to drive the two Sp-ASATs are from *S. lycopersicum* and the gene regulation system could be different between the two species. Second, the substrate pools in the trichomes could be different. For example, *S. pennellii* LA0716 produces more iso-C4 and iso-C10 containing acylsugars (Ning et al, *Plant Physiology*, 2015), while more iso-C5 and nC12 containing acylsugars are produced in *S. lycopersicum* (Ghosh et al, *Metabolomics*, 2013), suggesting the acyl-CoA pools in trichomes are different between the two species.

The biochemical assays conducted with recombinant Sp-ASAT2 and Sp-ASAT3 enzymes with data shown in Fig 3 provide strong evidence to show that the enzyme actions of Sp-ASAT2 and Sp-ASAT3 are reversed to those of *S. lycopersicum*.

- The figure legend in 3c is confusing to readers not familiar with these types of reactions: [“S” and “L” refer to short chain iC5-CoA or long chain nC12-CoA substrates, respectively]. The authors need to describe that enzyme assays use the acylated sugar acceptor and different CoA Esters.

We changed the figure to make it easier for the readers. S” and “L” were replaced with “iC5-CoA” and “nC12-CoA” in the revised figure. Corresponding changes were also made in the figure legends.

- In this context, how do the authors explain the perceived low levels of nC12-ester being produced by this enzyme compared with iC5-ester?
- Is it possible that a different acyltransferase is involved in this reaction? The results presented in this figure are qualitative? What would the quantitative differences be?

The results in this figure are qualitative with the goal to show how the P-type acylsugars are produced after flipping the enzyme reaction order given different acyl acceptor substrates[S1:5(iC5^{R4}) and S2:10(iC5^{R2}, iC5^{R4})]. When supplying 100 μM iC5-CoA or nC12-CoA as substrates, we observed lower triacylsucroses product peak areas with iC5 chain than with nC12 chain. We feel more comfortable not to compare this quantitative difference observed in vitro with the triacylsucroses produced in BIL6180, as differences in the acyl-CoA pools in type I/IV trichome tip cells is unknown.

- Given some of the promiscuity of some acyltransferases, is it not possible that a separate gene might be involved in generating longer chain esters? Perhaps the transgenic RNAi results back this possible role of Sp-ASAT3 and this should be stated more explicitly?

*In the introgression line IL11-3 that harbors Sp-ASAT3 but not Sp-ASAT2, no triacylsucrose is detected in trichome metabolites (Schilmiller et al, *Plant Cell*, 2015), suggesting no acyltransferases in that background could convert the Sp-ASAT3 product -- S2:10(iC5^{R2}, iC5^{R4}) -- to detectable P-type triacylsucroses.*

*However, we cannot rule out the possibility that a different acyltransferase in *S. pennellii* background is able to catalyze that reaction. RNAi suppression of *Sp-ASAT3* in *BIL6180* can't help answer this question, as it is designed to reduce the production of *S2:10(iC5^{R2},iC5^{R4})* in plant trichomes.*

- Another way to study this would be to look at enzymes extracted directly from trichomes to see how actively acylation would take place with native enzymes when the nC12-CoA the acyl acceptor was supplied to enzyme assays. While the evidence produced would be circumstantial, it might corroborate the low activities observed with recombinant proteins. It is remarkable that these types of studies with enzyme extracts from the plant rarely take place any more.

We agree that this could be a more direct way to characterize the acylsugar. However, there are two mitigating considerations in this case. First, based on the results from RNA-Seq (Ning et al, Plant Physiology, 2015) and GFP-promoter reporter lines (Schillmiller et al, Plant Cell, 2015; Fan et al, PNAS, 2016), the ASATs are only expressed in the single tip cells of type I/IV trichomes. Second, we would then need to remove the native acyl acceptor and acyl donor substrates without compromising the enzyme activities of these tiny amounts of enzymes.

The authors follow the recombinant enzyme studies with site specific mutagenesis studies to show that a few amino acids are likely to be responsible for the functional divergence of the F- and P-type acyl-sugars found in different tomato species found in different geographical areas of South America.

- While the data from mutagenesis studies provides evidence of the amino acid substitutions involved, the data is qualitative. It would be of interest to know more about the detailed kinetic changes that might be generated by these substitutions (Km, Vmax, Kcat)

Please see our response to the request for kinetic analysis by Reviewer 1.

Other comments/suggestions

1. Please explain the meaning of the following statement: “Line 185-87 [Published work demonstrated that *Sl-ASAT2* acylates the product of *Sl-ASAT1* and *Sp-ASAT1* — monoacylsucrose *S1:5 (iC5^{R4})* — at the R3 position to produce a diacylsucrose with both chains on the pyranose ring]. Do you mean [Published work demonstrated that *Sl-ASAT2* acylates monoacylsucrose *S1:5 (iC5^{R4})*, the product of *Sl-ASAT1* or *Sp-ASAT1* reactions at the R3 position to produce a diacylsucrose on the pyranose ring]?”

*The sentence was changed to “Published work demonstrated that *Sl-ASAT2* acylates the pyranose ring R3 position of the monoacylsucrose *S1:5 (iC5^{R4})* —the product of *Sl-ASAT1* and *Sp-ASAT1*—to produce a diacylsucrose.”*

2. Line 191-92: Meaning of: “Purification and NMR analysis of the diacylsucrose S2:10 (iC5, iC5R4) produced by the combination of SI”. Do you mean: “NMR analysis of the purified diacylsucrose S2:10 (iC5, iC5R4) produced by the combination of SI”

The reviewer’s explanation is more clearer and we made the changes accordingly in the revised manuscript.

3. Line 238-240: Meaning: “In vitro mutagenesis of C304G in SI-ASAT2 led to the ability to use S2:10 (iC5R2, iC5R4) and produce the P-type triacylsucrose S3:22 (iC5R2, nC12R3, iC5R4) (Fig. 4b).” Is this better: “In vitro mutagenesis of C304G in SI-ASAT2 led to the ability to convert S2:10 (iC5R2, iC5R4) to the P-type triacylsucrose S3:22 (iC5R2, nC12R3, iC5R4) (Fig. 4b).”

We agree with the reviewer’s suggestion and made the suggested change.

4. Line 329: “holding a unique position to inform us regarding the transition between F- and P- type acylsucrose pathways.”

The sentence was changed to “providing information regarding the evolutionary relationship of the F- and P- type acylsucroses pathways”

5. Lines 364-66: Meaning of “Evidence that these acylsucrose acyltransferases are sufficient for the P-type acylsucrose phenotype came from *S. lycopersicum* M82 transgenic lines expressing both Sp-ASAT2 and Sp-ASAT3 in type I/IV trichomes.” ‘Evidence that these acylsucrose acyltransferases are sufficient to account for the P-type acylsucrose phenotype came from *S. lycopersicum* M82 transgenic lines expressing both Sp-ASAT2 and Sp-ASAT3 in type I/IV trichomes.’

We agree with this suggestion and made the changes accordingly.

References:

Fan, P. et al. In vitro reconstruction and analysis of evolutionary variation of the tomato acylsucrose metabolic network. Proc. Natl. Acad. Sci. USA 113, E239–E248 (2016).

*Ghosh, B., Westbrook, T. C. & Jones, A. D. Comparative structural profiling of trichome specialized metabolites in tomato (*Solanum lycopersicum*) and *S. habrochaites*: acylsugar profiles revealed by UHPLC/MS and NMR. *Metabolomics* 10, 496–507 (2014).*

*Ning, J. et al. A feedback insensitive isopropylmalate synthase affects acylsugar composition in cultivated and wild tomato. *Plant Physiol.* 160, 1821–1835 (2015).*

*Schillmiller, A. L. et al. Functionally divergent alleles and duplicated loci encoding an acyltransferase contribute to acylsugar metabolite diversity in *Solanum* trichomes. *Plant Cell* 27, 1002–1017 (2015).*

We look forward to your feedback on this revised manuscript.

REVIEWERS' COMMENTS:

Reviewer #1 (Remarks to the Author):

All of my concerns have been addressed.

Reviewer #2 (Remarks to the Author):

The authors have addressed all my comments.

Reviewer #3 (Remarks to the Author):

The revised manuscript contains all of the revisions that were suggested in the earlier review of this paper. As indicated in the earlier review this is a very nice paper. I did find the authors took the easy way out when explaining that enzyme assays using proteins extracted directly from trichomes would be problematic since the enzymes are only found in single tip cells of type I/IV trichomes and since native acyl acceptor and acyl donor substrates would need to be removed without compromising the enzyme activities. Clearly there are well established methods for doing this, but unfortunately this is rarely performed anymore these days, since researchers assume that recombinant enzymes represent the situation with plant cell free extracted enzymes.